# `XIFBench`: Evaluating Large Language Models on Multilingual Instruction Following

**Zhenyu Li**[1]    **Xuefeng Bai**[1]*    **Yunfei Long**[2]    **Kehai Chen**[1]
**Yaoyin Zhang**[1]    **Xuchen Wei**[1]    **Juntao Li**[3]    **Min Zhang**[1]

[1]Harbin Institute of Technology, Shenzhen
[2]Queen Mary University of London
[3]Soochow University
`23s051032@stu.hit.edu.cn, chenkehai@hit.edu.cn, qp241311@qmul.ac.uk`

## Abstract

Large Language Models (LLMs) have demonstrated remarkable instruction-following capabilities across various applications. However, their performance in multilingual settings lacks systematic investigation, with existing evaluations lacking fine-grained constraint analysis across diverse linguistic contexts. We introduce `XIFBench`, a comprehensive constraint-based benchmark for evaluating multilingual instruction-following abilities of LLMs, comprising 558 instructions with 0-5 additional constraints across five categories (*Content*, *Style*, *Situation*, *Format*, and *Numerical*) in six languages spanning different resource levels. To support reliable and consistent cross-lingual evaluation, we implement three methodological innovations: cultural accessibility annotation, constraint-level translation validation, and requirement-based evaluation using English requirements as semantic anchors across languages. Extensive experiments with various LLMs not only quantify performance disparities across resource levels but also provide detailed insights into how language resources, constraint categories, instruction complexity, and cultural specificity influence multilingual instruction-following. Our code and data are available at `https://github.com/zhenyuli801/XIFBench`.

## 1  Introduction

Large Language Models (LLMs) have demonstrated remarkable capabilities in various natural language processing (NLP) tasks and real-world applications [1]. A significant driver of their widespread adoption is their instruction-following capability [2, 3], enabling them to understand and execute user commands while adhering to specified requirements [4, 5]. To support global applications and serve diverse linguistic communities, ensuring consistent multilingual instruction-following capability has attracted significant attention in advancing LLMs [6, 7].

While LLMs have shown promise in multilingual instruction-following, recent studies [7, 8] reveal significant performance disparities across languages (Figure 1). Factors such as the composition of multilingual instruction data have been investigated as potential contributors to this variation. However, existing evaluations often involve **relatively simple, direct instructions** and **coarse-grained evaluation methods**, such as pairwise comparison [9] or direct scoring [10]. These approaches fall short in capturing the complexity of real-world instructions and offer limited insight into how instruction-inherent factors affect cross-lingual performance.

**Constraint-based evaluations** have emerged as a promising alternative for assessing instruction-following [11, 12], as *constraints* are essential requirements within user instructions that LLMs should

---

*Corresponding author.

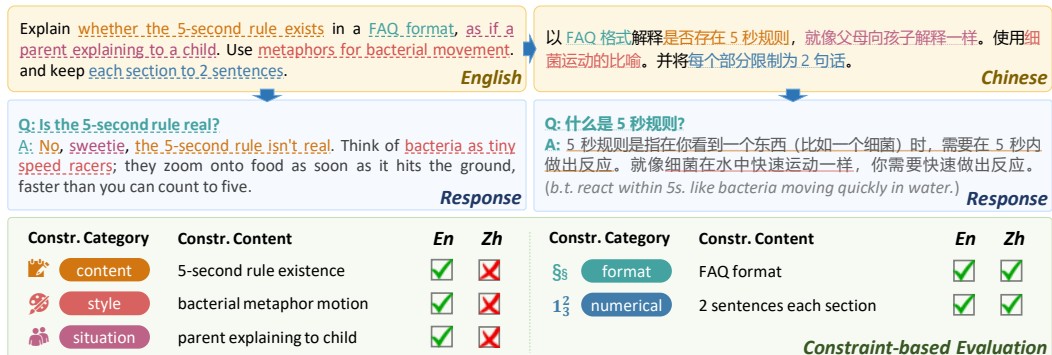

Figure 1: Discrepancies in LLMs' instruction-following across languages. Given English instruction and Chinese translation, Llama-3.1-8B response exhibits varied constraint-following, as shown in constraint-based evaluation. A concise back-translation (b.t.) of response is provided for reference.

satisfy. Constraints provides explicit targets for instruction formulation and evaluation, enabling a more fine-grained analysis of instruction-following capabilities. However, existing constraint-based frameworks are largely **limited to high-resource languages** like English and Chinese. This leaves a critical gap in understanding how multilingual properties, such as resource availability, influence LLMs' constraint-following ability in diverse linguistic contexts.

To bridge these critical gaps in multilingual instruction-following evaluation, we introduce XIFBench, a comprehensive benchmark designed to assess LLMs' capabilities with constraint-rich instructions across diverse linguistic contexts. XIFBench comprises 558 instructions, systematically augmented with 0-5 constraints drawn from a taxonomy covering five categories (*Content*, *Style*, *Situation*, *Format*, and *Numerical*) and 21 fine-grained dimensions. To ensure cross-lingual applicability, we specifically exclude language-dependent constraints. Beyond English, we translate these instructions into five languages (Chinese, Russian, Arabic, Hindi, and Swahili) representing high-, medium-, and low-resource settings. To enable insightful and reliable multilingual evaluation, we propose three methodological innovations: **(1)** We annotate the *cultural accessibility* of each instruction to investigate how these nuanced semantic elements impact cross-lingual instruction adherence, a factor often overlooked in existing evaluations. **(2)** We implement a constraint-level translation validation process that evaluates the semantic preservation of each constraint across languages. This ensures high fidelity and strict parallelism of constraints across languages. **(3)** We adapt the requirement-based evaluation methodology to multilingual contexts, using shared English requirements as semantic anchors. This approach ensures reliable and consistent instruction-following assessment across languages. Extensive experiments conducted on XIFBench with multiple LLMs not only quantify performance disparities across resource levels but also provide detailed, fine-grained insights into how factors such as language resources, constraint categories, instruction complexity, and cultural specificity influence multilingual instruction-following.

## 2   Related Work

**Instruction-Following Evaluation**   As the fundamental capability of LLMs to align with human intents, InstructGPT [2] evaluated instruction-following through human evaluation, employing direct scoring and binary adherence checks. Later works like AlpacaEval [9] and MT-Bench [10] utilized LLM judges for pairwise comparison and direct scoring, respectively. To assess LLMs on more complex tasks, WizardLM [13] and LIMA [14] introduced diverse and challenging instructions, but they still relied on coarse-grained evaluation methods.

The advancement of **constraint-based evaluation** has enabled fine-grained evaluation of instruction-following. CELLO [15] and CoDI-Eval [16] advanced this direction by collecting complex instructions and applying rule-based constraint checks. While IFEval [4] centers on objectively verifiable constraints, it predominantly covers format, numerical, and linguistic constraints. In contrast, Follow-Bench [11] expanded to more constraint types (e.g., content, situation, style) and analyzed the impact of constraint quantity with LLM-based evaluation. InfoBench [12] introduced requirement checklists for precise assessment and has been widely adopted in later studies. CFBench [17] developed a

comprehensive constraint taxonomy from real-world scenarios, and ComplexBench [18] further investigated the impact of constraint composition types. However, most studies primarily focus on English and Chinese, leaving a gap in understanding instruction-following across languages.

**Multilingual Instruction-Following Evaluation**  To evaluate multilingual instruction-following capabilities, existing work has primarily relied on translated versions of English-centric benchmarks, such as AlpacaEval [9], as demonstrated in various instruction tuning and evaluation efforts [19, 20, 7, 8]. Some research introduce language-specific benchmarks, such as Arabic MT-Bench [21] and CIFBench [22] for Chinese. Although Aya Evaluation Suite [23] provides native annotations and translations across 101 languages, its evaluation approaches remain coarse-grained.

While **constraint-based evaluation** is well-established for English, its multilingual exploration remains limited. Recent efforts have extended IFEval's framework to additional languages, like M-IFEval [24] covering English, French, Japanese and Spanish, and Multi-IF [5] extending to eight primarily Indo-European languages in multi-turn scenarios. However, these efforts largely target high- and medium-resource languages, thus offering limited insight into instruction-following capabilities across the full resource spectrum. Furthermore, by inheriting IFEval's focus on objective constraints (e.g., format, numerical), these benchmarks may fail to evaluate multilingual instruction-following in scenarios involving semantically rich constraints (e.g., style, situation), which are common in real-world instructions and potentially more sensitive to cross-lingual variations. In contrast, our work examines multilingual instruction-following across 6 languages spanning high, medium, and low resources, while including a comprehensive taxonomy of both objective and semantic constraints.

## 3 XIFBench Dataset

### 3.1 Dataset Construction

We construct XIFBench by preparing diverse seed instructions, then proceed with three automated stages: **(1)** augmenting them with constraints, **(2)** extracting fine-grained *evaluation requirements*, and **(3)** expanding the dataset to multiple languages. Figure 2 illustrates the construction workflow.

#### 3.1.1 Instruction Preparation

**Seed Instruction Selection**  To ensure the diversity and representativeness of XIFBench, we source evaluation instructions from AlpacaEval [9], WizardLM [13], and LIMA [14], which are widely used benchmarks for assessing multilingual instruction following capabilities [19, 20, 7, 8]. To prevent data leakage, we include only evaluation sets and omit all training data. To promote task diversity, we perform hierarchical clustering on these instructions, resulting in 131 distinct clusters. We then select one representative instruction from each cluster, ensuring a diverse and comprehensive instruction set. See Appendix C.1.1 for details.

**Instruction Filtering and Annotation**  While existing benchmarks provide a useful starting point, not all instructions are suitable for multilingual evaluation or further augmentation. Thus, we manually filter out ambiguous, overly difficult, or language-dependent instructions (e.g., *capitalized response*, *alliteration*), yielding a refined set of 106 instructions, named the **Easy Set**. Each instruction is further annotated with **cultural accessibility** labels to indicate culturally specific references (e.g., *Jesus Christ*, *YouTube*) that may not be universally understood across languages. Additionally, we decompose each instruction into *core instruction* and *input materials* to facilitate subsequent automated construction. See Appendix C.1.2 for details.

#### 3.1.2 Constraint Augmentation

**Constraint Taxonomy**  To systematically evaluate the impact of different constraint categories on multilingual instruction-following, we construct a taxonomy of 5 categories and 21 dimensions, building upon insights from prior constraint-based evaluations [11, 12, 17]. As illustrated in Figure 2, each constraint follows the structure *category - dimension - specification*, such as *content - include content - discuss scientific research*, guiding LLMs to include scientific research in their responses.

Our taxonomy covers five categories: **(1) Content**, which specifies what information should be included; **(2) Style**, which defines the tone and writing manners; **(3) Situation**, which describes

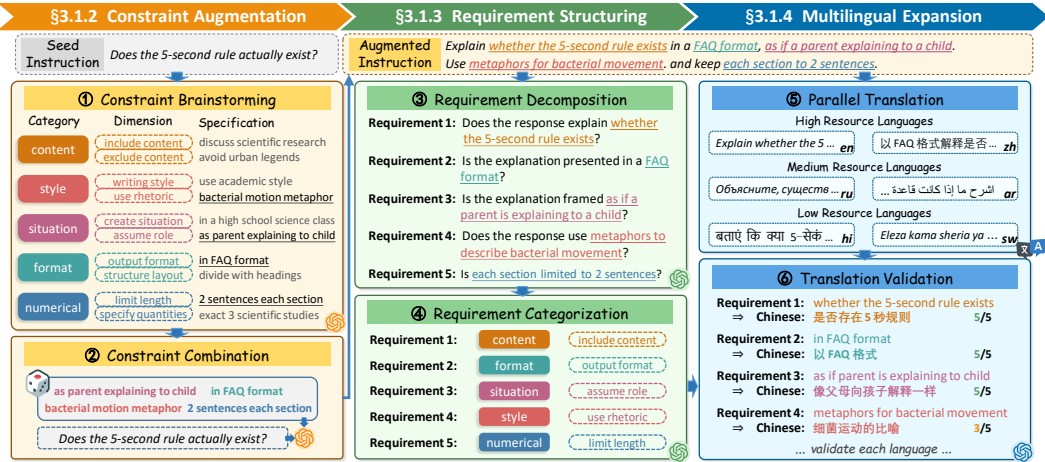

Figure 2: The automated pipeline for constructing XIFBench, consisting of three stages with six steps: Constraint Augmentation (§3.1.2), Requirement Structuring (§3.1.3), and Multilingual Expansion (§3.1.4). The example shown follows the same instruction as in Figure 1.

contextual settings such as assumed roles or environments; **(4) Format**, which outlines structural requirements for responses; and **(5) Numerical**, which involves quantitative factors like length or item counts. To ensure cross-lingual applicability, we intentionally exclude **language-dependent** constraints that depend on language-specific properties (e.g., *uppercase*, *alliteration*, *word count*), which were a focus in prior English- and Chinese-centric taxonomies. See Appendix C.2.1 for details.

**Constraint Brainstorming**   For each *core instruction*, we prompt GPT-4o [25][2] to generate multiple independent constraints across all 5 categories and 21 dimensions, ensuring that each dimension is adequately covered. This process results in a balanced constraint set while maximizing the variety of applicable constraints for each instruction. Additionally, it allows precise control over the types of constraints introduced in subsequent steps. See Appendix C.2.2 for details.

**Constraint Combination**   To balance instruction difficulty and diversity, we sample constraints from the brainstormed set to construct instructions with 1 to 5 additional constraints. The sampling algorithm prioritizes underrepresented categories for even distribution. We then prompt GPT-4o to integrate the selected constraints into fluent and natural instruction. This process yields the **Hard Set**, containing 530 *combined instructions*. See Appendix C.2.3 for details.

**Instruction Validation**   To ensure instruction quality, two human annotators independently assess each instruction in the **Hard Set** for **clarity** and **linguistic dependency** (i.e., whether it involves language-dependent constraints), and annotate its **cultural accessibility**. Instructions with defects and all their variations are removed, ensuring each **Easy Set** instruction has exactly five **Hard Set** counterparts with 1-5 added constraints. This validation yields 93 high-quality instructions in the Easy Set and 465 in the Hard Set. See Appendix C.2.4 for details.

### 3.1.3   Requirement Structuring

**Requirement Decomposition**   To enable fine-grained evaluation, and because these constraints are compound or altered during integration, we decompose instructions into atomic requirements, following prior works [12, 17, 18]. We complete this process by prompting GPT-4o to extract *evaluation requirements* from instructions in both Easy Set and Hard Set. As shown in Figure 2, each requirement is a binary (*YES/NO*) question (e.g., *Is each section limited to 2 sentences?*), ensuring a precise and interpretable assessment. See Appendix C.3.1 for details.

**Requirement Categorization**   Since the initial *core instructions* contain inherent constraints, and constraints from the combination step may be rephrased differently, we need to map the extracted *evaluation requirements* back to their original categories. To achieve this, we prompt GPT-4o to classify each requirement into the most suitable category and dimension within our predefined

---

[2]We use `gpt-4o-2024-08-06` for all constructions.

taxonomy. This automated process ensures consistency and facilitates a detailed analysis of the adherence to constraints in languages and models. See Appendix C.3.2 for details.

**Requirement Assessment**  To verify the accuracy of requirement extraction and categorization, we sample 10% of instructions (10 from the **Easy Set**, 50 from the **Hard Set**) and ask two annotators to independently evaluate the extracted requirements on four dimensions: **(1) Explicitness**: The requirement must be explicitly stated in the instruction. **(2) Completeness**: It must cover all explicitly stated constraints. **(3) Atomicity**: It should be atomic, addressing a single aspect. **(4) Categorization**: It should be assigned to the correct category. Our evaluation confirms their high quality, with at least 93.3% of requirements adhering to each criterion. See Appendix C.3.3 for details.

### 3.1.4  Multilingual Expansion

**Language Selection**  To ensure broad multilingual coverage while balancing evaluation costs, we select six representative languages—English, Chinese, Russian, Arabic, Hindi, and Swahili—based on prior multilingual instruction tuning works [7, 20]. These languages span different resource levels: **high** (English, Chinese), **medium** (Russian, Arabic), and **low** (Hindi, Swahili), following Joshi et al. [26]. The selection encompasses Indo-European (English, Russian, Hindi), Sino-Tibetan (Chinese), Afro-Asiatic (Arabic), and Niger-Congo (Swahili) language families, written in Latin, Cyrillic, Chinese, Arabic, and Devanagari scripts. This selection enables a systematic analysis of language resource impact on instruction following across diverse linguistic contexts.

**Parallel Translation**  We translate the *core instruction* of each English instruction in **Easy Set** and the *combined instruction* in **Hard Set** into five other languages using Google Translate, while keeping the *input materials* unchanged. This ensures that both code snippets and structured formats remain consistent across languages.

**Translation Validation**  While automatic translation is generally reliable, subtle semantic shifts can affect constraint-based evaluation. For example, the Chinese translation of "use metaphors for bacterial movement" in Figure 1 more closely resembles "use bacterial movement's metaphors." To validate such issues, as illustrated in Figure 2, we use GPT-4o and Google Translate's back-translation to evaluate the semantic preservation of each *evaluation requirement* across languages, ensuring that the translated instruction segment for each requirement aligns with its English counterpart. The aforementioned discrepancy is successfully identified and rated as 3/5, indicating borderline preservation. Our analysis shows that less than 1.4% of *evaluation requirements* exhibit inconsistencies across languages. Human annotations on a sampled subset confirm this trend and show moderate agreement with automatic scores, confirming that the translation quality is sufficient for reliable multilingual evaluation. See Appendix C.4 for details.

### 3.2  Dataset Statistics

| Dataset | #Inst. | #Req. | Avg. Tokens | Avg. Req. |
|---------|--------|-------|-------------|-----------|
| Easy Set | 93 | 269 | 36.3 ± 53.0 | 2.9 ± 1.1 |
| Hard Set | 465 | 1395 | 69.4 ± 56.0 | 5.01 ± 1.7 |
| Overall | 558 | 1664 | 48.1 ± 22.4 | 4.66 ± 1.8 |

Table 1: Dataset statistics of `XIFBench`, including instance count, evaluation requirements count, and average token lengths.

Table 1 summarizes `XIFBench`, which contains 558 instructions and 1,664 *evaluation requirements*. Each instruction averages 48.1 ± 22.4 tokens, including *core instructions* and optional *input materials*. To enable multilingual evaluation, instructions are translated into five languages (Chinese, Russian, Arabic, Hindi, and Swahili), yielding 3,348 instances. The *evaluation requirements* remain unchanged across languages for consistent evaluation. See Appendix D for details.

### 3.3  Evaluation Protocol

**Evaluation Framework**  We adopt the widely used LLM-based evaluation protocol from InfoBench [12], using predefined *evaluation requirements* to assess how well the responses follow

instructions. For every instruction and its model-generated response, we prompt an LLM to generate a concise observation and a binary *YES/NO* decision for each requirement.

While originally designed for English-only or Chinese-only settings, this protocol raises an overlooked question when applied to multilingual evaluation: **how does the language of *evaluation requirements* affect evaluation reliability and consistency?** Translating requirements into the same language as the response may seem more natural and contextually aligned, but it may lead to unreliable assessments due to mistranslations and reduce cross-lingual consistency without shared criteria.

To address this, we retain the original English *evaluation requirements* across all languages, using them as fixed semantic anchors. Specifically, we provide the original English instruction, *evaluation requirements*, and metadata (categories and dimensions) for each instance, ensuring a shared and stable evaluation framework. This avoids translation-induced inaccuracies and ensures cross-linguistic comparability. See Appendix E for details.

**Evaluation Metrics**   We define two primary metrics: **Requirement Following Rate (RFR)** and **Instruction Following Rate (IFR)**. **RFR** measures the percentage of evaluation requirements correctly satisfied across all instructions, offering a fine-grained view of adherence to specific constraints. **IFR** quantifies the percentage of instructions where *all* requirements are met, providing a stricter assessment of overall compliance. These metrics capture LLMs' ability to follow instructions across constraints and complexities.

**Metrics Formulation**   Let $\mathcal{I}^{(l)}$ be the set of instructions in language $l \in \{\text{en}, \ldots, \text{sw}\}$, each instruction $i^{(l)} \in \mathcal{I}^{(l)}$ has an associated set of *evaluation requirements* $\mathcal{R}_i$, identical across languages. Given an LLM-generated response $o_i^{(l)}$ to instruction $i^{(l)}$, the adherence to a requirement $r \in \mathcal{R}_i$ is denoted as $e_{i,r}^{(l)}$, a binary value: 1 if satisfied, and 0 otherwise. The LLM-based evaluation function $\mathcal{E}$ determines requirement adherence:

$$e_{i,r}^{(l)} = \mathcal{E}(i^{(l)}, o_i^{(l)}, i^{(\text{en})}, r), \tag{1}$$

where $\mathcal{E}$ returns 1 if the requirement is met, 0 otherwise. English instructions are simplified to:

$$e_{i,r}^{(\text{en})} = \mathcal{E}(i^{(\text{en})}, o_i^{(\text{en})}, r). \tag{2}$$

The **RFR** for language $l$ is the proportion of satisfied requirements over all instructions:

$$\text{RFR}^{(l)} = \frac{\sum_{i^{(l)} \in \mathcal{I}^{(l)}} \sum_{r \in \mathcal{R}_i} e_{i,r}^{(l)}}{\sum_{i^{(l)} \in \mathcal{I}^{(l)}} |\mathcal{R}_i|}. \tag{3}$$

The **IFR** is the proportion of instructions where all requirements are satisfied:

$$\text{IFR}^{(l)} = \frac{1}{|\mathcal{I}^{(l)}|} \sum_{i^{(l)} \in \mathcal{I}^{(l)}} \prod_{r \in \mathcal{R}_i} e_{i,r}^{(l)}. \tag{4}$$

## 4   Experiments

### 4.1   Automatic Evaluation

We employ GPT-4o[3] as the evaluation judge to assess 9 LLMs across two categories: **(1) Closed-source LLMs**: GPT-4o[4], Gemini-2.0-Flash [27], and Claude-3.5-Sonnet [28]; **(2) Open-source LLMs**: Llama-3.1-70B, Llama-3.1-8B [29], Qwen-2.5-72B, Qwen-2.5-14B, Qwen-2.5-7B [30], and GLM-4-9B-Chat [31]. We access closed-source models via their official APIs, while open-source models run locally using vLLM [32] for efficient inference. We use greedy decoding with a maximum token limit of 4096 to ensure determinism and prevent truncation, keeping all other hyperparameters at default values. LLMs are tested on both sets. We report **RFR** and **IFR** scores for each LLM.

Table 2 presents the main results of the automatic evaluation. **From a language resource perspective**, we observe that: **(1)** Performance correlates with resource levels. High-resource languages (e.g.,

---

[3]We use `gpt-4o-2024-08-06` for all evaluations.

[4]We assess `gpt-4o-2024-11-20` to avoid self-evaluation.

| Metric | Requirement Following Rate (RFR) | | | | | | | Instruction Following Rate (IFR) | | | | | | |
|---|---|---|---|---|---|---|---|---|---|---|---|---|---|---|
| Language | En | Zh | Ru | Ar | Hi | Sw | Avg. | En | Zh | Ru | Ar | Hi | Sw | Avg. |
| *Closed-Source Language Models* | | | | | | | | | | | | | | |
| GPT-4o | **93.6** | 92.5 | 92.7 | 90.8 | **92.8** | 90.8 | **92.2** | 76.9 | 73.3 | 74.2 | 69.2 | **73.8** | 65.6 | 72.2 |
| Gemini-2.0-Flash | 93.3 | **93.0** | **93.0** | 91.9 | 92.0 | 89.5 | 92.1 | **78.1** | 76.7 | 77.0 | 75.8 | 71.7 | **69.2** | 74.7 |
| Claude-3.5-Sonnet | 89.1 | 81.3 | 84.6 | 75.9 | 76.1 | 74.5 | 80.2 | 66.1 | 53.0 | 61.6 | 46.1 | 43.5 | 40.1 | 51.8 |
| Avg. (Closed) | 92.0 | 88.9 | 90.1 | 86.2 | 87.0 | 84.9 | **88.2** | 73.7 | 67.7 | 70.9 | 63.7 | 63.0 | 58.3 | **66.2** |
| *Open-Source Language Models* | | | | | | | | | | | | | | |
| Llama-3.1-70 | 91.7 | 83.4 | 87.3 | 76.4 | 80.9 | 73.4 | 82.2 | 70.9 | 48.9 | 58.2 | 40.2 | 42.7 | 34.8 | 49.3 |
| Qwen-2.5-72B | 90.5 | 89.1 | 89.2 | 85.4 | 82.3 | 40.9 | 79.6 | 67.7 | 63.3 | 64.9 | 57.2 | 51.3 | 10.4 | 52.5 |
| Qwen-2.5-14B | 89.7 | 88.5 | 87.8 | 82.1 | 70.0 | 23.5 | 73.6 | 64.7 | 61.1 | 60.6 | 51.4 | 33.9 | 5.0 | 46.1 |
| Glm-4-9B | 86.4 | 87.2 | 85.0 | 78.6 | 71.3 | 25.6 | 72.4 | 56.5 | 60.3 | 54.2 | 43.2 | 34.6 | 4.1 | 42.2 |
| Llama-3.1-8B | 87.6 | 79.1 | 79.5 | 63.2 | 66.8 | 38.6 | 69.1 | 58.9 | 42.8 | 44.9 | 25.3 | 27.2 | 9.7 | 34.8 |
| Qwen-2.5-7B | 87.8 | 87.4 | 83.3 | 77.1 | 59.6 | 10.0 | 67.6 | 59.9 | 57.3 | 52.3 | 41.0 | 21.1 | 1.1 | 38.8 |
| Avg. (Open) | 89.0 | 85.8 | 85.4 | 77.1 | 71.8 | 35.3 | 74.1 | 63.1 | 55.6 | 55.9 | 43.1 | 35.1 | 10.9 | 44.0 |

Table 2: Following rates (%) of different models on XIFBench. The highest rates among *closed-source models* are in **bold**, while those among *open-source models* are underlined. The **Avg.** column represents the average rate across all languages for each model. The **Avg. (Closed)** and **Avg. (Open)** rows represent the average rate for models within their respective categories.

English, Chinese) exhibit stable performance, medium-resource ones (e.g., Russian, Arabic) show greater variance, and low-resource languages (e.g., Hindi, Swahili) experience steep IFR declines, sometimes nearing zero. **(2)** The RFR-IFR gap is most pronounced in low-resource languages. Models achieve reasonable RFR scores but struggle with IFR, indicating difficulties in full instruction adherence. For example, in Hindi, several models exceed 80% in RFR but fall below 40% in IFR. **(3)** Instruction adherence remains a challenge across all languages. The persistent RFR-IFR gap suggests that limitations extend beyond data scarcity—models can follow individual constraints but struggle with holistic instruction execution.

**From a model capability perspective**, we observe in Table 2 that: **(1)** Closed-source models excel in RFR but still face IFR challenges. Despite the strong performance, models like GPT-4o and Gemini-2.0-Flash achieve RFR above 90% but IFR only around 70%, revealing limitations in full adherence. **(2)** Open-source models exhibit sharper IFR declines. While their RFR remains stable in high-resource languages, it degrades significantly in lower-resource settings, indicating weaker cross-lingual generalization. **(3)** Larger models generally outperform smaller ones, though exceptions (e.g., Qwen-2.5-14B in Swahili) likely stem from data composition rather than scaling limitations.

## 4.2 Agreement Evaluation

Considering cross-lingual variation in LLM evaluator agreement [33], we assess our protocol's reliability and consistency through agreement studies. We compare three settings: **(1) Direct Scoring (DS)** [10], which holistically rates responses from 1 to 10, considering dimensions such as helpfulness and relevance. Following [11, 12], scores above 5 are considered to meet all *evaluation requirements*; **(2) Ours w/ Trans. Reqs.**, where *evaluation requirements* are translated into the instruction's language, simulating evaluation without a shared semantic anchor to test concerns raised in 3.3; and **(3) Ours w/ Eng. Reqs.**, our proposed protocol using shared English *evaluation requirements*.

We randomly sample 1080 *instruction-response* pairs across six languages, covering both the **Easy Set** and **Hard Set**. Responses are generated by three LLMs with varying scales: Gemini-2.0-Flash, Qwen-2.5-72B, and Glm-4-9B. GPT-4o performs evaluations for all settings. Two human annotators assess the alignment between GPT-4o's evaluation and human judgement for each *evaluation requirement*. We report agreement rates between GPT-4o and human annotations. See Appendix F.2 for details.

As shown in Table 3, our protocol (**Ours w/ Eng. Reqs.**) outperforms DS across languages, with a slight drop in low-resource ones. Its high average agreement (94.7%) and low standard deviation (1.6) confirm its reliability and cross-lingual consistency. The high DS agreement in Swahili doesn't reflect greater reliability, but rather results from poor model performance–often scoring below 5–coincidentally aligning with human labels. Notably, compared to using translated requirements (**Ours w/ Trans. Reqs.**), our protocol (**Ours w/ Eng. Reqs.**) achieves higher agreement across languages

| Methods | En | Zh | Ru | Ar | Hi | Sw | Avg. | Std. |
|---|---|---|---|---|---|---|---|---|
| DS | 71.7 | 56.7 | 53.9 | 55.0 | 58.3 | 69.4 | 60.7 | 6.5 |
| Ours w/ Trans. Reqs. | 93.7 | 89.1 | 89.1 | 86.7 | 84.6 | 84.9 | 88.5 | 3.1 |
| Ours w/ Eng. Reqs. | **95.9** | **96.7** | **95.0** | **93.0** | **95.5** | **92.5** | **94.7** | **1.6** |

Table 3: Agreement rates between GPT-4o and human annotations across different languages (%) under **DS** and our protocol variants. **Bold** indicates best results.

(Avg: 94.7% vs. 88.5%), indicating improved reliability. Moreover, it exhibits lower cross-lingual variance (Std: 1.58 vs. 3.10), demonstrating enhanced consistency and fairer comparisons. These results support our initial concern that translating evaluation requirements can compromise both reliability and cross-lingual consistency.

## 5 Analysis

### 5.1 How Do Constraint Categories Affect Cross-lingual Performance?

We only use the **Hard Set** for its balanced constraint distribution and focus on **RFR** to align with the granularity of constraint analysis. We analyze three models spanning different capabilities: Gemini-2.0-Flash, Qwen-2.5-72B, and Glm-4-9B. Other settings follow Section 4.1.

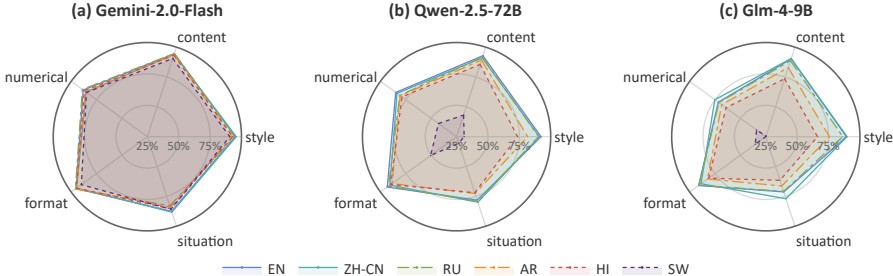

Figure 3: Cross-lingual RFR performance across constraint categories for three representative models. Each radar chart illustrates the RFR scores across different languages within each constraint category.

As illustrated in Figure 3, we find that **higher-capacity models exhibit more robust cross-lingual constraint adherence**. **(1)** For high-capacity models like Gemini-2.0-Flash, the radar plots form nearly regular polygons across languages, indicating stable adherence with minimal cross-lingual variation. **(2)** In contrast, mid- and low-capacity models (Qwen-2.5-72B, Glm-4-9B) exhibit progressive degradation in polygon size and regularity as language resources decrease. This degradation is most pronounced for style and situation constraints, suggesting these constraints are more sensitive to language-specific factors.

Further examining constraint categories, we observe that **some constraints exhibit universal properties, while others are language-dependent**. **(1)** Format and numerical constraints are more resilient to language variations, as indicated by denser polygon edges (i.e., more consistent performance). This suggests they rely on universal linguistic properties. **(2)** In contrast, style, situation, and content constraints are more sensitive to language resources, with sparser polygon edges as resources decline. Style and situation constraints degrade the most, reinforcing their strong dependence on language-specific properties, while content constraints show moderate degradation.

### 5.2 How Does Instruction Complexity Impact Cross-lingual Performance?

We use both the **Easy Set** (no additional constraints) and **Hard Set** (augmented with 1-5 constraints per instruction) to cover a wide range of instruction complexity. **IFR** is used to align with instruction-level analysis. Model selection and other experimental settings follow Section 5.1.

As shown in Figure 4, we find that **stronger LLMs maintain more consistent performance across instruction complexity levels**. **(1)** Gemini-2.0-Flash demonstrates stable performance across all language resource levels without additional constraints and exhibits relatively smooth degradation

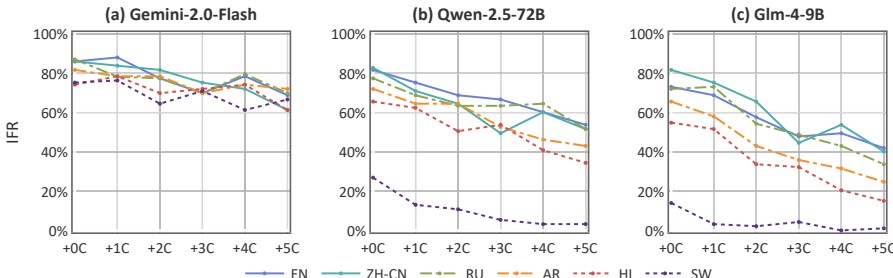

Figure 4: Cross-lingual IFR performance across instruction complexity levels for three representative models. Each group shows IFR scores across languages per additional constraint count (+xC).

as constraints increase, maintaining acceptable IFR even with five constraints. **(2)** The mid- and low-capacity models (Qwen-2.5-72B, Glm-4-9B) not only perform slightly worse but also degrade more steeply across languages, dropping to unacceptable IFR levels with five constraints in Swahili.

Further analysis reveals that **performance degradation under complex instructions is not clearly correlated with language resource levels**. **(1)** Across all languages, IFR tends to decline approximately linearly as instruction complexity increases, with relatively similar slopes regardless of resource level. **(2)** Swahili appears to degrade more steadily in Qwen-2.5-72B and Glm-4-9B, but this is due to its already low baseline performance. Interestingly, we find that degradation patterns vary across models and languages in non-trivial ways. We provide further analysis in Appendix G.3.

### 5.3 To What Extent Does Cultural Specificity Influence Instruction Following?

We use the **Easy Set** and **Hard Set** to compare LLM performance on culturally universal and specific instructions (Section 3.1). Both **RFR** and **IFR** are included for a comprehensive view of instruction following. Model selection and other settings follow Section 5.1.

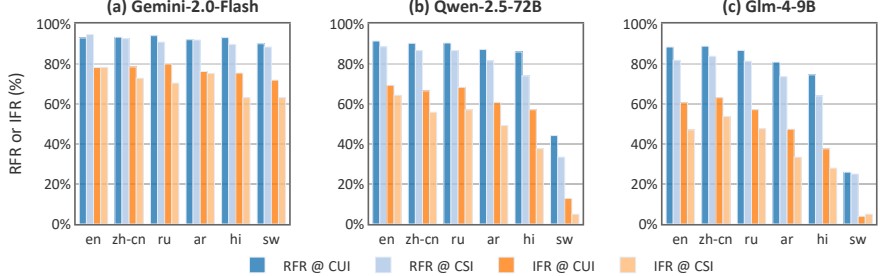

Figure 5: Cross-lingual RFR and IFR performance of culturally universal (CUI) and specific (CSI) instructions across three models. Each group presents following rates per type across languages.

As shown in Figure 5, we observe several key findings: **(1)** As language resources decrease, both RFR and IFR decline for culturally universal and specific instructions. The degradation rate is slightly steeper for culturally specific instructions but not substantially. **(2)** More capable models like Gemini-2.0-Flash show smaller performance gaps between culturally universal and specific instructions, while smaller models like GLM-4-9B degrade more on culturally specific instructions. **(3)** IFR is more sensitive to cultural specificity than RFR, especially as language resources or model capacity decrease, since failures in cultural constraints hinder complete instruction following. These findings suggest that despite pre-training exposure to cultural content, cultural specificity still impacts instruction following, particularly for smaller models and low-resource languages.

## 6 Conclusion

In this paper, we introduced `XIFBench`, a comprehensive constraint-based benchmark for evaluating the multilingual instruction-following capabilities of LLMs. Featuring a systematic design with constraint-rich instructions across six languages and three methodological innovations for reliable

cross-lingual assessment, `XIFBench` enables fine-grained analysis. Through extensive experiments, we quantified performance disparities across resource levels and provided detailed insights into the influence of language resources, constraint categories, instruction complexity, and cultural specificity on multilingual instruction-following. In summary, `XIFBench` provides a valuable resource and analytical tool for advancing research and development of robust multilingual LLMs.

## Acknowledgements

This work was supported in part by the National Natural Science Foundation of China (62276077, 62406091, 62350710797, U23B2055), in part by the Guangdong Basic and Applied Basic Research Foundation (2024A1515011205), and in part by the Shenzhen Science and Technology Program (KQTD20240729102154066, ZDSYS20230626091203008).

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

# A    Limitations

While XIFBench provides a comprehensive evaluation framework, we acknowledge several opportunities for future extensions:

- Our benchmark relies on LLMs for construction and evaluation, supplemented by careful human verification. However, this pipeline may still be susceptible to biases inherent in the "LLM-evaluate-LLM" process. We encourage future work to explore fully manual or alternative human-in-the-loop methodologies for benchmark development.

- Since our benchmark relies on parallel translations, the cultural accessibility annotations are primarily derived from English contexts, which may not fully capture the cultural diversity across different languages. Directly incorporating annotations from native speakers for non-English instructions would provide a more comprehensive cross-cultural perspective.

- Our benchmark focuses on general-purpose LLMs, excluding reasoning-specialized models (e.g., o1, DeepSeek-R1, Gemini-2.5-Pro-Preview) due to budget and API constraints. These models, which often infer implicit constraints or explore alternative solutions in their thoughts, may exhibit different patterns in constraint adherence.

- Another interesting scenario of multilingual instruction-following is code-mixing, i.e., the use of multiple languages in a single instruction. While our benchmark could theoretically evaluate code-mixing by keeping *input materials* untranslated, we do not include this in our current study. This is because *input materials* contain diverse content types and would compromise the interpretability of our findings. We leave this as future work.

- We identify key factors affecting multilingual instruction following, but translating these insights into concrete model improvements remains an open challenge. Future work could explore how our findings can inform training strategies to enhance multilingual instruction following, particularly for semantic-rich constraints and low-resource languages.

# B    Broader Impacts

XIFBench is a benchmark designed to evaluate the multilingual instruction-following capabilities of LLMs. By providing a comprehensive evaluation framework, it aims to advance research in this area and promote the development of more robust and capable LLMs. Regarding its broader social impacts, the benchmark can facilitate the development of LLMs that are better equipped to understand and follow instructions in multiple languages, potentially improving accessibility and usability for non-English speakers. This could lead to more inclusive AI systems that cater to a diverse user base. However, a potential negative impact is that developers might prioritize improving model performance on the more widely-spoken languages included in the benchmark, potentially widening the gap between high-resource and low-resource languages.

# C    Dataset Construction Details

In this part, we provide detailed information on the construction of XIFBench, including instruction preparation, constraint augmentation, requirement structuring, and multilingual expansion.

## C.1    Instruction Preparation Details

### C.1.1    Seed Instruction Selection

As described in Section 3.1.1, we source instructions from AlpacaEval, WizardLM, and LIMA via their official repositories on Hugging Face and GitHub. Since WizardLM and LIMA primarily contribute instruction-tuning datasets along with evaluation sets, we use only the evaluation sets to construct our benchmark, thereby avoiding potential data leakage.

For hierarchical clustering, we utilize the best-performing model, `all-mpnet-base-v2`, in Sentence Transformers[5], to encode the instructions and compute the Euclidean distances between them. We

---

[5]`https://www.sbert.net/`

then apply hierarchical clustering to group similar instructions. To determine an appropriate number of clusters, we iterate over various threshold distances and select a value that ensures both diversity and sufficient seed instructions. Ultimately, we set the threshold distance to 1.8, resulting in 131 clusters. To maximize diversity, we randomly select a single instruction from each cluster.

### C.1.2 Instruction Filtering and Annotation

In this section, we provide detailed information on the filtering and annotation process for the selected seed instructions, as described in Section 3.1.1. The authors select data by following the annotation guidelines in Table 4. Annotators are instructed to assign binary *true/false* values to five dimensions specified in the guideline, with optional comments.

The *clarity* dimension measures the understandability of the instruction, while *linguistic independence* ensures the instruction's language-neutral properties. *Cultural accessibility* is also annotated to assess dependence on specific references that may not be globally accessible or understandable. Additionally, we impose restrictions of fewer than five explicit constraints and four content constraints to filter out instructions that might become overly complex after augmentation. This process results in an **Easy Set** containing 106 instructions.

The annotation guidelines for the decomposition of *core instructions* and *input materials* are provided in Table 5. Here, *core instructions* refer to the essential tasks described in the instructions, while *input materials* serve as supporting information for these tasks. This annotation process is also conducted by the authors.

### C.2 Constraint Augmentation Details

### C.2.1 Constraint Taxonomy

The constraint taxonomy from Section 3.1.2 is presented in Table 6. The examples provided are brainstormed constraints for the instruction *Does the 5-second rule exist?*, as illustrated in Figure 2.

### C.2.2 Constraint Brainstorming

The prompt used for constraint brainstorming, as described in Section 3.1.2, is presented in Table 7. We provide both the *core instruction* and *input materials* to GPT-4o, instructing it to generate 12 constraints for each category. We set the maximum generation length to 2048 tokens and use a temperature of 0 to ensure stable results.

### C.2.3 Constraint Combination

The prompt used for constraint combination, as described in Section 3.1.2, is shown in Table 8. For each instruction, we sample five times, incrementally selecting 1 to 5 constraints. To ensure category diversity and balance, only unsampled categories are considered at each step. Given a *core instruction* and the sampled constraints, we prompt GPT-4o to generate five distinct combined instructions using these sampled constraints. The maximum generation length is set to 512 tokens, and a temperature of 0 is used to ensure stable results.

### C.2.4 Instruction Valiadation

As described in Section 3.1.2, we conducted a manual annotation to evaluate the quality of the *combined instructions* in the **Hard Set**. The annotation process followed the guidelines in Table 4, except for the two constraint-related criteria. Two annotators independently assessed all instructions for *clarity* and *linguistic independence*, and annotated *cultural accessibility*. To measure inter-annotator agreement, we randomly selected 10% of the Hard Set (50 instances) and included them in both annotators' assignments. Disagreements in this subset were resolved conservatively: if either annotator marked an instruction as defective or culturally specific, it was classified as such. Both annotators were graduate students in computer science and received prior training on the annotation guidelines. We report the proportion of instructions identified as clear, linguistically independent, and culturally universal, along with inter-annotator agreement scores.

As shown in Table 9, nearly all instructions are clear and linguistically independent, demonstrating the high quality of the dataset and the reliability of the automatic construction workflow. The strong

inter-annotator agreement further supports the consistency of the annotation process. However, the agreement on cultural accessibility is lower, likely due to the subjective nature of this criterion.

| Criteria | Ratio (%) | Agreement (%) |
|---|---|---|
| Clarity | 96.6 | 96.0 |
| Linguistic Independence | 100.0 | 100.0 |
| Cultural Accessibility | 70.6 | 80.0 |

Table 9: Evaluation results for various criteria, showing the **Ratio** of instructions meeting each criterion and the **Agreement** (%) between evaluators.

Following the annotation, we removed all unclear and linguistically dependent instructions, along with their corresponding variants in both the Easy Set and the Hard Set, ensuring that each instruction retains all variants across 0 to 5 additional constraints. This process also affected a small number of culturally specific instructions. As a result, the proportion of culturally universal instructions in the Hard Set increased slightly from 70.6% to 70.8%, with a total of 329 culturally universal instructions remaining.

Moreover, to assess whether the constraint augmentation process introduces additional culture-specific content, we measured cultural accessibility across both the Easy and Hard Sets. We found that 31.2% of Easy Set instructions and 29.2% of Hard Set instructions are culturally specific. Notably, 97.6% of instructions retain the same cultural accessibility label after augmentation–only 2.4% introduced new culture-specific constraints. This suggests that the constraint combination process has minimal impact on cultural accessibility.

## C.3 Requirement Structuring Details

### C.3.1 Requirement Decomposition

The prompt used for requirement decomposition, as described in Section 3.1.3, is presented in Table 10. We use GPT-4o to decompose each *core instruction* in the Easy Set and each *combined instruction* in the Hard Set into multiple *evaluation requirements* for assessing LLM-generated responses. In our prompt, we explicitly instruct GPT-4o to generate a list of requirements that: **(1)** are strictly derived from the given instruction without inference, **(2)** are atomic and indivisible to enable fine-grained evaluation, and **(3)** are formatted as *YES/NO* questions for binary assessment. To ensure stable results, we set the maximum generation length to 512 tokens and use a temperature of 0.

One might argue that the brainstormed constraints could also serve as evaluation requirements. However, this does not eliminate the necessity of requirement decomposition. The need for decomposition arises not merely from the need to format them as questions, but due to follows: **(1)** *Evaluation requirements* are essential for assessing the Easy Set instructions as well as the corresponding components within the Hard Set instructions. **(2)** The constraints generated by GPT-4o are sometimes compound statements encompassing multiple requirements. For example, a constraint like *"Maintain a neutral but informative tone"* actually consists of two distinct aspects: neutrality and informativeness. Furthermore, during the constraint combination step, these constraints may be rephrased or semantically altered, making explicit requirement decomposition necessary for accurate evaluation.

### C.3.2 Requirement Categorization

The prompt used for requirement categorization in Section 3.1.3 is presented in Table 11. We provide GPT-4o with a list of *evaluation requirements* along with their instructions and ask it to map each requirement to the predefined constraint taxonomy for further analysis. The maximum generation length is set to 256 tokens, with a temperature of 0.

### C.3.3 Requirement Assessment

We present the details of the requirement assessment included in Section 3.1.3. For each *evaluation requirement*, we assess its quality from four dimensions: **explicitness**, **completeness**, **atomicity**, and **categorization**. A detailed explanation of each dimension is provided in the annotation guideline (Table 12), resulting in five binary evaluation criteria for each requirement. The **categorization**

dimension is assessed from two perspectives: **(1)** whether the requirement is correctly categorized and **(2)** whether it is assigned to the correct dimension. The **completeness** dimension is evaluated at the instruction level, while the other three dimensions are assessed at the requirement level.

To ensure the reliability of the assessment, two annotators independently annotated a randomly sampled subset, comprising 10% of the instructions from both sets (10 from the Easy Set and 50 from the Hard Set). We calculate both the proportion of high-quality requirements and the inter-annotator agreement on this subset. Disagreements were resolved using a conservative resolution strategy, similar to the approach described in Section C.2.4.

| Criteria | Ratio (%) | Agreement (%) |
|---|---|---|
| Explicitness | 99.8 | 99.9 |
| Completeness | 93.3 | 95.0 |
| Atomicity | 95.9 | 96.6 |
| Categorization | 94.8 | 96.3 |
| Dimension | 94.0 | 95.5 |

Table 13: Evaluation results for various criteria, presenting the **Ratio** of requirements or instructions (for completeness) that meet each criterion and the **Agreement** (%) between evaluators.

As shown in Table 13, almost all requirements adhere to the explicitness criterion, indicating that very few nonexistent requirements were generated. Additionally, the majority of *evaluation requirements* are rated as complete, atomic, and correctly categorized in both constraint category and dimension. These results demonstrate the high quality of the generated requirements. Furthermore, the inter-annotator agreement scores confirm the consistency and reliability of the annotation process.

### C.4 Multilingual Expansion Details

In this section, we detail the translation validation process introduced in Section 3.1.4. Since manual annotation of the entire dataset for Chinese, Russian, Arabic, Hindi, and Swahili is costly and time-consuming, we leverage GPT-4o and Google Translate to automatically assess translation quality. Our validation procedure follows a requirement-based evaluation framework, ensuring that each instruction and its translation are analyzed for both accuracy and consistency.

Specifically, for each instruction-translation pair, we prompt GPT-4o to identify corresponding segments in the English instruction and its translated counterpart, using predefined *evaluation requirements* as reference points. Additionally, we incorporate back-translation via Google Translate to mitigate potential biases, particularly for low-resource languages. The LLM annotator then provides a concise assessment of translation accuracy and assigns a quality score on a 1-5 scale (1 = Failed Preservation, 5 = Perfect Preservation), where a score of 3 serves as the threshold for acceptable translation quality. The evaluation prompt is provided in Table 14.

| Score | Zh | Ru | Ar | Hi | Sw |
|---|---|---|---|---|---|
| 5 | 93.5 | 95.1 | 93.3 | 94.3 | 89.5 |
| 4 | 5.3 | 3.3 | 4.4 | 4.3 | 7.6 |
| 3 | 0.9 | 1.0 | 1.0 | 0.8 | 1.6 |
| 2 | 0.3 | 0.5 | 0.9 | 0.4 | 1.0 |
| 1 | 0.0 | 0.1 | 0.5 | 0.2 | 0.3 |

(a) Distribution of translation quality scores (%) from GPT-4o on the full dataset.

| Score | Zh | Ru | Ar | Hi | Sw |
|---|---|---|---|---|---|
| 5 | 81.5 | 79.5 | 83.0 | 76.8 | 79.3 |
| 4 | 13.3 | 15.9 | 12.0 | 17.8 | 15.9 |
| 3 | 4.2 | 3.3 | 2.7 | 4.3 | 3.7 |
| 2 | 1.0 | 0.8 | 2.3 | 0.8 | 1.0 |
| 1 | 0.0 | 0.4 | 0.0 | 0.2 | 0.0 |

(b) Distribution of translation quality scores (%) from human annotation on 20% of the dataset.

Figure 6: Comparison of translation quality score distributions across languages. **Left:** GPT-4o annotations on the full dataset. **Right:** Human annotations on a 10% subset.

To quantify translation quality, we analyze the distribution of *evaluation requirement* scores across different languages. To further validate the automatic evaluation and assess its agreement with human judgments, we randomly sample 20% of English instructions (18 from the Easy Set and 90 from the Hard Set) and their translations in all non-English languages. For each language, two native-speaking annotators independently scores the sampled translations on a 1-5 scale, following guidelines slightly

adapted from Table 14 to facilitate human annotation. Additionally, the segments extracted by GPT-4o during scoring are provided to annotators to help identify the relevant portions for each requirement. We take the lower of the two scores for each translation to conservatively estimate translation quality.

The results, summarized in Table 6a, show that the vast majority of translations receive high-quality ratings. Notably, the proportion of requirements scoring below 3 is low across all languages, with Arabic exhibiting the highest at only 1.4%. These findings suggest that the translations are largely accurate and consistent with the original English instructions, making them suitable for inclusion in the benchmark without significantly affecting overall evaluation outcomes.

Human annotation results on the sampled subset, shown in Table 6b, further support this conclusion: over 97.7% of all translations are rated ≥3. Disagreements occur in about 25% of segments, but are mostly minor (e.g., human = 4 vs. GPT-4o = 5), indicating that GPT-4o may slightly overestimate quality. These results validate our automatic scoring as a reasonable proxy for translation quality and support the reliability of our multilingual benchmark.

# D    Dataset Statistics Details

As illustrated in Figure 7a, the distribution of *evaluation requirements* in the Easy Set is largely dominated by **Content** constraints, reflecting a lack of diversity in constraint types. However, despite being constructed based on the Easy Set, the Hard Set exhibits a more balanced distribution across all categories, indicating a diverse and well-balanced constraint augmentation process.

Further analysis of the requirement count distribution across the Easy and Hard Sets is shown in Figure 7b. While both sets follow a similar uniform distribution, the Hard Set exhibits a broader range of requirement counts, with a pronounced peak at 5 requirements, compared to 2 in the Easy Set. This distribution highlights the greater diversity and complexity of the Hard Set, making it a more challenging evaluation benchmark for LLMs.

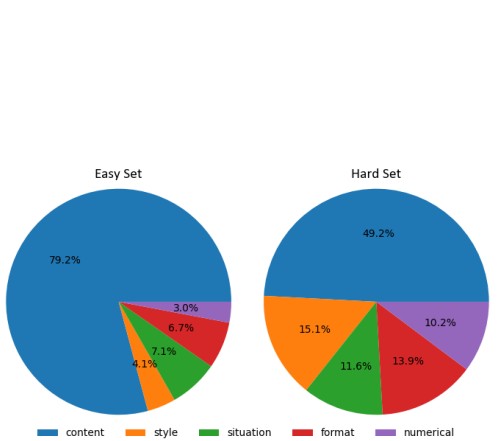

(a) Requirement category distribution across the Easy Set and Hard Set.

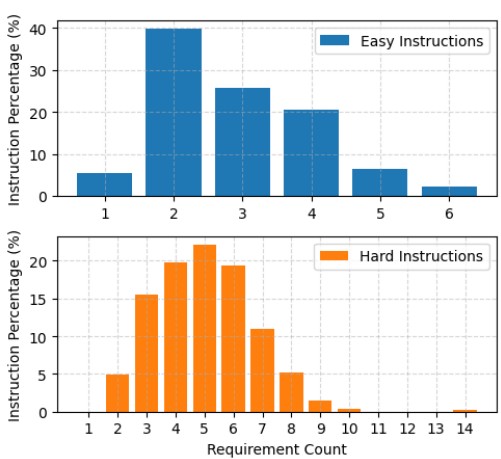

(b) Requirement count distribution across the Easy Set and Hard Set.

Figure 7: Distribution analysis of requirements in the dataset. **Left:** Category distribution across Easy and Hard sets. **Right:** Count distribution across Easy and Hard sets.

# E    Evaluation Protocol Details

The prompt used in the evaluation framework (Section 3.3) is provided in Table 15 for English instructions and Table 16 for other languages. For English instructions, GPT-4o receives the instruction, the corresponding response, and the *evaluation requirements* to assess the response's adherence to the given instruction. The model first provides observations on the response, followed by a binary *YES/NO* decision for each requirement, indicating whether the response meets it. For instructions in other languages, we follow a similar procedure, providing GPT-4o with the translated instruction, the

corresponding response, the original English instruction, and the *evaluation requirements*. To ensure stable results, we set the maximum generation length to 2048 tokens and use a temperature of 0.

# F    Experiment Details

In this section, we provide detailed information about the experiments discussed in Section 4, including supplemental details on the automatic evaluation of LLMs, and detailed description of agreement between GPT-4o and human annotations.

## F.1    Automatic Evaluation

The official API names of the models used in Section 4.1 are: gpt-4o-2024-11-20, gemini-2.0-flash-001 and claude-3-5-sonnet-20241022. The open-source models and their repositories on Hugging Face[6] are: THUDM/glm-4-9b-chat, meta-llama/Llama-3.1-8B-Instruct, meta-llama/Llama-3.1-70B-Instruct, Qwen/Qwen2.5-7B-Instruct, Qwen/Qwen2.5-14B-Instruct, and Qwen/Qwen2.5-72B-Instruct. For the inference of open-source models, we used at most 2 H20-80G GPUs for inference. Evaluating an LLM on the entire XIFBench dataset using gpt-4o-2024-08-06 (without batching) costs approximately \$24.

## F.2    Agreement Evaluation

In this section, we detail the process of evaluating agreement between GPT-4o's assessments and human annotations across the three evaluation settings outlined in Section 4.2.

- **Direct Scoring (DS)**: GPT-4o scores each response on a scale from 1 to 10. This approach does not explicitly assess whether each constraint is satisfied, but rather offers a holistic judgment of the response quality. Following prior work [11, 12], responses scoring above 5 are considered to meet all *evaluation requirements* for calculating agreement against our human-labeled requirement-level ground truth. We adopt the evaluation prompt from Zheng et al. [10] without modification.
- **Ours w/ Trans. Reqs.** (Ours with Translated Requirements): GPT-4o evaluates responses using *evaluation requirements* translated into the same language as the instruction. For example, a Chinese *instruction-response* pair was evaluated with Chinese-translated *evaluation requirements* and metadata (categories, dimensions). This setting simulates a multilingual evaluation scenario without a shared semantic anchor, an issue raised in Section 3.3, allowing us to examine how the language of evaluation criteria affects reliability and consistency.
- **Ours w/ Eng. Reqs.** (Ours with English Requirements): This is our proposed evaluation protocol. GPT-4o evaluates all *instruction-response* pairs using the original English *evaluation requirements*, regardless of the language of the instruction. For instance, a Swahili *instruction-response* pair is evaluated using English *evaluation requirements* and metadata. This ensures a consistent semantic anchor across languages, avoiding translation-induced inaccuracies and ensuring cross-linguistic comparability.

We use Google Translate to obtain translated requirements and translated metadata for the **Ours w/ Trans. Reqs.** setting. The evaluation prompt template (see Table 15) is adapted by replacing language-specific parts—such as the descriptive text "English instruction" and its corresponding placeholder {*english_instruction*}—with their language-specific equivalents (e.g., "Chinese instruction" and {*chinese_instruction*}).

To construct the evaluation set, we randomly sampled 10% of the English instructions (10 from the **Easy Set** and 50 from the **Hard Set**) as a representative subset, resulting in 60 English instructions with 295 individual *evaluation requirements*. Including their translations across the six languages yields 360 instructions. To ensure diversity in response quality, we collected outputs from three representative LLMs with varying capabilities: Gemini-2.0-Flash, Qwen-2.5-72B, and GLM-4-9B. This results in 1080 *instruction-response* pairs in total.

To obtain ground-truth labels, two human annotators assess whether GPT-4o's binary decisions align with human judgment for each evaluation requirement, following the guidelines in Table 17.

---

[6] https://huggingface.co/models

Specifically, annotators are shown GPT-4o's generated observations (from the **Ours w/ Eng. Reqs.** setting) to assist in identifying relevant segments of the response, though they are instructed not to base their decisions solely on them. Annotators then determine whether they agree with GPT-4o's binary decision (*YES/NO*) on each requirement. In cases of disagreement, we reverse GPT-4o's decision to reflect the human judgment. We report the agreement rates between the human-verified ground truth labels and GPT-4o across the three evaluation settings.

# G  Extended Experimental Results and Analysis

In this section, we present additional experimental results and analyses that support the reliability of our evaluations and provide extended insights into multilingual instruction-following capabilities across diverse languages.

## G.1  Assessing the Influence of Translation Quality on Evaluation Results

While Section 3.1.4 and Appendix C.4 demonstrate that `XIFBench`'s translation quality is generally acceptable, there remains a possibility that subtle translation defects might affect evaluation reliability. To address this concern, we assess whether translation quality impacts evaluation results.

We leverage the translation quality scores from Section 3.1.4, which measure how well each segment in translated instruction preserves the semantics of its corresponding English *evaluation requirement*. In our 1-5 scale, scores below 3 indicate failed semantic preservation, 3 represents the borderline acceptance threshold, scores above 3 indicate sufficient preservation, and 5 represents perfect preservation. To quantify the potential impact, we simulate three evaluation scenarios:

- **Eval. w/ All Reqs.**: We consider evaluation decisions for all requirements across languages, regardless of translation quality scores. This corresponds to our standard evaluation approach used throughout the paper, such as the main results in Table 2.

- **Eval. w/ Acceptable Reqs.**: We only consider evaluation decisions for requirements with translation quality scores $\geq 3$ (meeting or exceeding our acceptance threshold). This simulates results if defective translations were excluded from evaluation.

- **Eval. w/ Perfect Reqs.**: We only consider evaluation decisions for requirements with perfect translation quality scores (5). This simulates an ideal scenario with perfect translations.

| Metric | Requirement Following Rate (RFR) | | | | | Instruction Following Rate (IFR) | | | | |
|---|---|---|---|---|---|---|---|---|---|---|
| Language | Zh | Ru | Ar | Hi | Sw | Zh | Ru | Ar | Hi | Sw |
| *Gemini-2.0-Flash* | | | | | | | | | | |
| Eval. w/ All Reqs. | 93.0 | 93.0 | 91.9 | 92.0 | 89.5 | 76.7 | 77.0 | 75.8 | 71.7 | 69.2 |
| Eval. w/ Acceptable Reqs. | 93.0 | 93.1 | 92.3 | 92.2 | 89.6 | 76.7 | 77.6 | 76.3 | 72.6 | 69.2 |
| Eval. w/ Perfect Reqs. | 93.1 | 93.2 | 92.9 | 92.3 | 89.7 | 77.4 | 77.9 | 76.9 | 73.1 | 70.6 |
| Std. | 0.04 | 0.10 | 0.51 | 0.15 | 0.14 | 0.41 | 0.45 | 0.54 | 0.72 | 0.83 |
| *Qwen-2.5-72B* | | | | | | | | | | |
| Eval. w/ All Reqs. | 89.1 | 89.2 | 85.4 | 82.3 | 40.9 | 63.3 | 64.9 | 57.2 | 51.3 | 10.4 |
| Eval. w/ Acceptable Reqs. | 89.1 | 89.2 | 85.9 | 82.6 | 41.0 | 63.4 | 64.9 | 58.1 | 52.3 | 10.8 |
| Eval. w/ Perfect Reqs. | 89.2 | 89.5 | 86.6 | 82.6 | 41.6 | 64.2 | 65.8 | 59.3 | 53.0 | 11.5 |
| Std. | 0.08 | 0.17 | 0.57 | 0.18 | 0.34 | 0.47 | 0.52 | 1.08 | 0.90 | 0.55 |
| *GLM-4-9B* | | | | | | | | | | |
| Eval. w/ All Reqs. | 87.2 | 85.0 | 78.6 | 71.3 | 25.6 | 60.3 | 54.2 | 43.2 | 34.6 | 4.1 |
| Eval. w/ Acceptable Reqs. | 87.3 | 85.1 | 78.9 | 71.6 | 25.7 | 60.6 | 55.0 | 43.7 | 35.0 | 4.5 |
| Eval. w/ Perfect Reqs. | 87.4 | 85.3 | 79.5 | 71.8 | 26.1 | 61.0 | 55.7 | 44.6 | 35.7 | 5.0 |
| Std. | 0.07 | 0.15 | 0.43 | 0.25 | 0.27 | 0.36 | 0.72 | 0.73 | 0.55 | 0.45 |

Table 18: Following rates (%) across different evaluation settings. **Eval. w/ All Reqs.** considers all requirements regardless of translation quality. **Eval. w/ Acceptable Reqs.** only considers requirements with translation quality scores $\geq 3$. **Eval. w/ Perfect Reqs.** only considers requirements with perfect translation quality scores (5). **Std.** shows the standard deviation across the three settings.

We follow the model selction and other experimental settings used in 5.1 to report RFR and IFR scores for these three scenarios across non-English languages for three representative models: Gemini-2.0-Flash, Qwen-2.5-72B, and GLM-4-9B. The results are shown in Table 18.

As shown in Table 18, the results for RFR and IFR indicate that better translation quality leads to slightly higher performance scores. However, the standard deviation across scenarios remains consistently below 0.57% for RFR and below 1.08% for IFR, indicating minimal impact on overall evaluation outcomes. Importantly, the relative ranking across languages remains stable across all filtering scenarios, confirming that our conclusions regarding multilingual instruction-following capabilities are valid and robust to translation quality variations.

## G.2 Assessing the Impact of LLM Evaluator Choice on Human Agreement

Our main evaluation protocol relies on GPT-4o as the sole automatic evaluator. A valid concern is that this choice could introduce model-specific biases, potentially affecting the reliability of our results. To investigate this and assess the robustness of our evaluation setup, we conduct additional experiments comparing the performance of several distinct LLM evaluators against human judgments.

We leverage the human annotations from Section 4.2 as the ground truth for this analysis. We compare the agreement rates of the original evaluator (GPT-4o) with three other diverse LLMs. They are:

- **GPT-4.1**: To assess consistency and potential variations within the same advanced model family as the original evaluator.
- **Gemini-2.5-Flash**: A strong, cost-efficient model from a different developer (Google) to provide an external perspective.
- **DeepSeek-R1**: A leading open-source model known for its strong reasoning capabilities, representing another distinct model architecture and training philosophy.

We use the same experimental setup as in Section 4.2, calculating the human agreement rate for each of these models across all six evaluated languages. The results are summarized in Table 19.

| Model Judge | En | Zh | Ru | Ar | Hi | Sw |
|---|---|---|---|---|---|---|
| GPT-4o | **95.9** | **96.7** | **95.0** | **93.0** | **95.5** | **92.5** |
| GPT-4.1 | 91.9 | 91.6 | 87.4 | 89.1 | 90.5 | 84.6 |
| Gemini-2.5-Flash | 90.2 | 91.9 | 85.3 | 87.0 | 89.5 | 86.0 |
| DeepSeek-R1 | 90.9 | 90.2 | 80.0 | 82.8 | 84.2 | 87.0 |

Table 19: Human agreement rates (%) for different LLM evaluators. We use the human annotations from Section 4.2 as ground truth. **GPT-4o** (our paper's default) shows the highest agreement.

From the results in Table 19, we observe several key findings: **(1)** The original evaluator, **GPT-4o**, achieves the highest agreement with human judgments. This is the expected upper-bound performance, given that the evaluation prompt was iteratively refined for this model during development. **(2)** Among the alternative judges, models like **GPT-4.1** and **Gemini-2.5-Flash** maintain relatively high and stable agreement rates across languages (e.g., all scores >85%). In contrast, the open-source **DeepSeek-R1** exhibits significantly higher variance, performing comparably in English (90.9%) but dropping substantially in languages like Russian (80.0%) and Arabic (82.8%). **(3)** This analysis **validates the robustness of our evaluation method, particularly for high-resource languages** like English and Chinese, where all judges show >90% agreement. More importantly, it **highlights a critical challenge for the field**: the inconsistent reliability of alternative LLM judges in mid- and low-resource languages. This demonstrates that while our setup is sound, substituting the evaluator is not a trivial step and requires careful, per-language validation.

## G.3 Revisting the Impact of Instruction Complexity on Multilingual Instruction Following

While Section 5.2 provides an analysis of the impact of instruction complexity on multilingual instruction-following capabilities, two additional aspects warrant further exploration:

- Beyond observing the expected decline in fully following (IFR) as constraint count increases, RFR can provide complementary insights into how LLMs handle complex instructions.

- Although multilingual instruction-following degradation under complex instructions was not clearly correlated with language resource levels in our initial analysis, a more detailed examination might reveal subtle patterns.

To investigate these aspects, we conduct additional slope analysis on cross-lingual performance from both RFR and IFR perspectives. Following the model selection and experimental settings used in Section 5.2, we evaluate three representative models: Gemini-2.0-Flash, Qwen-2.5-72B, and GLM-4-9B. For each language, we calculate RFR and IFR across different complexity levels (measured by the count of additional constraints, +xC), as depicted in Figure 4. We then fit a linear regression model to quantify the rate of performance change as complexity increases, where the x-axis represents the number of additional constraints and the y-axis represents either RFR or IFR. The resulting slopes (percentage points per complexity level) are reported in Table 20.

| Metric | Requirement Following Rate (RFR) Slopes | | | | | | | Instruction Following Rate (IFR) Slopes | | | | | | |
|---|---|---|---|---|---|---|---|---|---|---|---|---|---|---|
| Language | En | Zh | Ru | Ar | Hi | Sw | Std. | En | Zh | Ru | Ar | Hi | Sw | Std. |
| Gemini-2.0-Flash | +0.05 | -0.56 | +0.15 | +0.08 | +1.06 | +0.75 | 0.57 | -3.49 | -4.73 | -2.56 | -2.00 | -2.15 | -2.33 | 1.04 |
| Qwen-2.5-72B | -0.41 | -0.47 | +0.23 | -0.41 | -0.58 | -1.54 | 0.61 | -5.35 | -5.81 | -4.06 | -6.05 | -6.21 | -4.37 | 0.89 |
| GLM-4-9B | -0.83 | -1.33 | -0.95 | -0.50 | -0.59 | -0.12 | 0.42 | -6.39 | -8.37 | -8.21 | -8.32 | -8.40 | -2.06 | 2.54 |

Table 20: Slopes of performance change (% per complexity level) across instruction complexity levels for different languages. Positive slopes indicate performance improvement with increasing complexity, while negative slopes indicate degradation. **Std.** represents the standard deviation of slopes across all languages.

**For RFR slopes**, we observe two key findings: **(1)** Compared to IFR slopes where the minimum absolute value is 2.00, RFR slopes stay relatively close to zero and stable across complexity levels. This indicates that models often fail to fully satisfy all constraints simultaneously (declining IFR) but consistently satisfy a similar proportion of individual constraints (stable RFR) even as complexity increases. **(2)** Comparing slopes across languages, we find minimal variations regardless of resource level, with standard deviations (0.42-0.61) consistently smaller than those for IFR (0.89-2.54). This suggests that the phenomenon of "satisfying a fixed proportion of constraints" is minimally affected by language resource availability.

**For IFR slopes**, we observe more pronounced patterns: **(1)** All slopes are below -2.00, confirming significant performance degradation as complexity increases, consistent with the visualization in Figure 4. **(2)** Cross-linguistic comparison reveals model-specific patterns not initially apparent:

- GLM-4-9B exhibits a clear two-tier degradation pattern: English degrades moderately (-6.39), while other languages except Swahili degrade much more severely (approximately -8.3), revealing a notable advantage for the highest-resource language.

- Qwen-2.5-72B shows partial correlation with language resource levels: high-resource languages (English and Chinese) degrade less moderately than medium-resource (Arabic) and low-resource (Hindi) languages.

- Gemini-2.0-Flash interestingly reverses this trend: high-resource languages (English and Chinese) show steeper degradation than other languages, possibly because they start from higher baseline performance, allowing more room for decline.

It's important to note that for Qwen-2.5-72B and GLM-4-9B, Swahili IFR rapidly approaches zero even at lower complexity levels. This leads to artificially shallow slopes for Swahili that should be interpreted cautiously, as they simply reflect the lack of further room for degradation rather than resilience to complexity.

### G.4 Multilingual Instruction Following Performance of Open-Source Multilingual Models

Our main paper (Section 4.1) primarily evaluates large-scale, general-purpose LLMs. A complementary analysis is to investigate the performance of dedicated open-source multilingual models. These models are specifically pre-trained on a wide array of languages and provide a different perspective on multilingual instruction-following.

To this end, we evaluated several open-source multilingual models of comparable size: **CohereLabs/aya-23-8B** (8B, 23 languages), **utter-project/EuroLLM-9B-Instruct** (9B, 35 languages), **CohereLabs/aya-101** (13B, 101 languages), and **bigscience/bloomz-7b1** (7B, 46 languages).[7] We followed the inference and evaluation setup described in Section 4.1. For models lacking official user-assistant prompting templates (i.e., aya-101 and bloomz-7b1), we used customized system prompts and simulated dialogue formatting to avoid instruction continuation issues.

| Metric | Requirement Following Rate (RFR) | | | | | | | Instruction Following Rate (IFR) | | | | | | |
|---|---|---|---|---|---|---|---|---|---|---|---|---|---|---|
| Language | En | Zh | Ru | Ar | Hi | Sw | Avg. | En | Zh | Ru | Ar | Hi | Sw | Avg. |
| aya-23-8B | 79.3 | 78.3 | 78.7 | 77.1 | 72.6 | 4.7 | 65.1 | 41.8 | 41.0 | 43.5 | 39.5 | 31.7 | 1.5 | 33.2 |
| EuroLLM-9B | 73.2 | 74.5 | 70.4 | 62.3 | 58.4 | 8.8 | 57.9 | 35.7 | 38.4 | 32.4 | 23.8 | 21.2 | 1.8 | 25.6 |
| aya-101 | 16.6 | 13.0 | 15.7 | 14.8 | 13.3 | 15.1 | 14.8 | 3.4 | 3.1 | 3.1 | 2.9 | 2.5 | 3.4 | 3.1 |
| bloomz-7b1 | 8.1 | 6.3 | 4.2 | 7.1 | 6.4 | 3.3 | 5.9 | 2.0 | 1.8 | 1.6 | 2.3 | 2.3 | 0.9 | 1.8 |

Table 21: Performance of open-source multilingual models on `XIFBench`. All models were evaluated under the same setup as in Section 4.1. The results highlight the critical role of instruction tuning.

The results are presented in Table 21. We observe two primary findings: **(1)** Among the instruction-tuned models, **Aya-23-8B** consistently outperforms **EuroLLM-9B-Instruct** across all languages except Swahili. This may be attributable to its more balanced multilingual training data, whereas EuroLLM-9B appears to be more English and Western-European centric. **(2)** Both **aya-101** and **bloomz-7b1** demonstrate significantly lower performance on both RFR and IFR metrics. This is likely because neither model has undergone dedicated user-assistant instruction tuning, which is critical for adhering to the types of complex directives present in `XIFBench`. These results suggest that while multilingual pre-training is a necessary foundation, dedicated instruction tuning remains a dominant factor for achieving strong multilingual instruction-following capabilities.

### G.5 Investigating the "Carry-Over Effect" from Source Benchmarks

A potential concern regarding `XIFBench` is the "carry-over effect," where the instruction-following capabilities measured might simply reflect those from the source benchmarks used to construct our benchmark (e.g., LIMA, AlpacaEval, and WizardLM). If models' performance on our newly constructed Hard Set is highly correlated with their performance on the Easy Set (which is derived from these sources), it might suggest that the Hard Set does not introduce novel challenges, but merely replicates the source benchmarks.

To investigate this, we compute the Pearson correlation coefficient between model scores on the Easy Set and their corresponding scores on the Hard Set. The analysis is granular: for each of the three source benchmarks (AlpacaEval, WizardLM, LIMA) and each of the six languages, we correlate the performance of all evaluated models on the Easy subset with their performance on the Hard subset. The results are presented in Table 22.

| Metric | RFR Pearson Correlation | | | | | | IFR Pearson Correlation | | | | | |
|---|---|---|---|---|---|---|---|---|---|---|---|---|
| Source | En | Zh-cn | Ru | Ar | Hi | Sw | En | Zh-cn | Ru | Ar | Hi | Sw |
| AlpacaEval | 0.69 | 0.91 | 0.72 | 0.97 | 0.95 | 0.98 | 0.88 | 0.87 | 0.81 | 0.92 | 0.95 | 0.97 |
| WizardLM | 0.49 | 0.76 | 0.73 | 0.84 | 0.73 | 0.97 | 0.68 | 0.66 | 0.78 | 0.93 | 0.58 | 0.89 |
| LIMA | 0.70 | 0.93 | 0.73 | 0.86 | 0.58 | 0.99 | 0.76 | 0.87 | 0.85 | 0.75 | 0.64 | 0.95 |
| **Avg.** | 0.63 | 0.87 | 0.73 | 0.89 | 0.75 | 0.98 | 0.77 | 0.80 | 0.81 | 0.87 | 0.72 | 0.94 |

Table 22: Pearson correlation of model performance on the Easy Set versus the Hard Set. Each cell represents the correlation across all evaluated models for a specific language (column) and instructions derived from a specific **source benchmark** (row).

The results reveal distinct patterns. **(1) For English**, the correlations are moderate (e.g., 0.63 RFR Avg., 0.77 IFR Avg.). This is a favorable outcome, indicating that while the Hard Set tests related

---

[7]We also attempted to evaluate `google/Gemma-3-12B-it` but encountered unresolved inference issues (cache length mismatching) during our testing, so it was excluded from this analysis.

skills, it is not redundant. The construction process, which was centered on English instructions, successfully introduced novel challenges that alter model rankings.

**(2) For non-English languages**, correlations are significantly higher, particularly in low-resource languages (e.g., Swahili, 0.98 RFR Avg.). We hypothesize this is due to models' weaker overall capabilities in these languages, which "compresses" their performance into a narrow range. With a small initial performance gap, relative model rankings remain stable even on the Hard Set, as fundamental linguistic competence is the dominant factor across both sets. This results in highly consistent rankings (high correlation).

**(3) Crucially, high correlation does not imply a lack of challenge.** This stability in relative rankings is coupled with significant *absolute* performance drops. For instance, in the LIMA-Swahili subset, Qwen-2.5-72B's score drops from 40.0 RFR and 10.5 IFR on the Easy Set to 31.0 RFR and 2.1 IFR on the Hard Set. This demonstrates that even when relative model rankings are stable (high correlation), the absolute difficulty introduced by the Hard Set is substantial.

Therefore, we conclude that the Hard Set introduces meaningful new challenges. The moderate correlation in English shows the introduction of novel task properties, while the high correlation in other languages–coupled with large performance drops–indicates that the Hard Set effectively functions as a more difficult test of the same fundamental capabilities, not a redundant one.

**Instruction Validation Task**

**Task Introduction**
Your task is to validate each instruction against specific rules given in the criteria section.
We will use instructions that meet these criteria as seed instructions to create a multilingual instruction-following benchmark for Large Language Models (LLMs). We will complexify these seed instructions by adding constraints to create a hard set of instructions for LLMs. We will translate these instructions into different languages such as `English`, `Chinese`, `Russian`, `Arabic`, `Hindi`, `Swahili`, and more.

**Input & Output Explanation**
{{INPUT_OUTPUT_EXPLAINATION}}

**Validation Criteria**
Please check each instruction against the following criteria step-by-step:

**Rule 1: Is the instruction clear to understand? [YES/NO]**
**Definition:** The instruction should be clear to understand, avoiding ambiguity or complexity. This ensures the instruction can be understood by LLMs with no confusion.
**Positive Example:** "What is the meaning of life?"
*Accepted Reason:* The instruction is clear to understand.
**Negative Example:** "Summarize this website LOTTADIGITAL.COM"
*Rejected Reason:* The instruction is ambiguous due to the lack of context of the website.
{{...}}

**Rule 2: Does the instruction maintain linguistic independence across different languages? [YES/NO]**
**Definition:** The instruction should be language-neutral, avoiding any features specific to particular languages. This ensures the instruction can be applied across different languages without linguistic barriers.
Linguistic features are language-specific elements such as unique alphabets, wordplay, puns, grammatical structures, or punctuation requirements that may not be universally applicable.
**Positive Example:** "Write a paragraph about black holes."
*Accepted Reason:* The instruction is linguistically independent and can be applied across various languages.
**Negative Example:** "Write a paragraph about black holes using all vowels (A, E, I, O, U)."
*Rejected Reason:* The instruction relies on English-specific features (vowels), making it unsuitable for other language systems.
{{...}}

**Rule 3: Does the instruction maintain cultural accessibility across different languages? [YES/NO]**
**Definition:** The instruction should focus on universal concepts while avoiding culturally specific references that may not be globally accessible or understood. This ensures the instruction can be applied across different languages without cultural barriers.
Cultural references are elements that either have limited geographic/linguistic spread (such as regional platforms, products, customs, or policies) or require specific cultural background knowledge (such as historical, religious, or traditional contexts). These elements may not be universally accessible or meaningful across different cultural settings.
**Positive Example:** "Why did humans evolve to believe in gods?"
*Accepted Reason:* The instruction discusses a universal human phenomenon.
**Negative Example:** "Why did humans evolve to believe in Jesus Christ?"
*Rejected Reason:* The instruction references a specific religious figure, making it less universally accessible.
{{...}}

**Rule 4: Does the instruction contain equal to or fewer than 5 explicit constraints? [YES/NO]**
**Definition:** The instruction should contain $\leq$ 5 explicit constraints. This ensures the instruction is not too complex after adding constraints.
Constraints are specific requirements that LLMs must follow in their responses. These are categorized into five types: content, style, situation, format, and numerical constraints.
**Positive Example:** "Write a short story about a detective solving a mystery."
*Accepted Reason:* The instruction contains 2 constraints: format (short story) and content (detective solving mystery).
**Negative Example:** "Write a short story about a detective solving a mystery. Include a twist at the end. Use descriptive language. Make sure the characters are well-developed. Restrict the story to 500 words."
*Rejected Reason:* The instruction contains more than 5 constraints across content (detective solving mystery, well-developed characters), style (twist at the end, descriptive language), format (short story), and numerical (500 words).
{{...}}

**Rule 5: Does the instruction contain equal to or fewer than 4 content constraints? [YES/NO]**
**Definition:** The instruction should contain $\leq$ 4 content constraints. This ensures the instruction is balanced and not overly focused on content aspects after adding constraints.
Content constraints specify the thematic or topical focus of a response, detailing what information should be included or excluded to guide the depth and scope of the content.
**Positive Example:** "Recommend three science fiction books suitable for teenagers."
*Accepted Reason:* The instruction contains 2 content constraints: science fiction genre and teenage suitability.
**Negative Example:** "Recommend three science fiction books suitable for teenagers. Include a brief summary of each book. Provide the publication year for each book. Mention the author of each book."
*Rejected Reason:* The instruction contains more than 4 content constraints including the genre, audience, summaries, dates, and authors.
{{...}}

**Task Workflow**
1. Read the `prompt` field.
2. Evaluate the instruction against each rule.
3. If feeling uncertain, you can (recommended from most to least encouraged):
   - Use web search or external LLMs to understand the instruction better.
   - Test any LLMs with the instruction to see if it produces the expected output.
4. Fill in the `human_annotation_tags` fields based on your evaluation.
5. Add any additional comments or notes in the `notes` field if necessary.

Table 4: Human Annotation Guidelines for Instruction Validation Task

**Instruction Decomposition Task**

**Task Introduction**

Your task is to decompose each given instruction into `input_material` and `core_instruction`.

We will use these decomposed core instructions to create a multilingual instruction-following benchmark for Large Language Models (LLMs). When evaluating LLMs' performance on this benchmark, we will also provide the input materials alongside their corresponding instructions.

**Input Explanation**

{{INPUT_EXPLANATION}}

**Output Explanation**

You need to decompose the `prompt` into two components:

- `input_material`: The source content or reference materials that require processing or analysis as part of the task. These materials contain no directives or requirements.

- `core_instruction`: The primary directive or task to be completed, including task requirements, scenarios, formats, styles, and other relevant information.

**Task Guidelines**

After reading the full instruction in the `prompt` field, separate it into `core_instruction` and `input_materials` by copying the exact text without modifications. Consider the following guidelines:

1. The `core_instruction` should retain its complete semantic meaning even when the input materials are removed.

2. `input_materials` are typically independent components that can be separated from the instruction without altering its overall meaning.

3. In many cases, `input_materials` are preceded by two newline characters ("\n\n"). While this can be a helpful indicator, it is not a strict rule.

4. If there are no clear `input_materials`, consider the entire text as the core instruction.

**Decomposition Examples**

**Example 1**

**Given Instruction:**

"What is the meaning of life?"

**Decomposition:**

- **Input Materials:** None

- **Core Instruction:** "What is the meaning of life?"

**Explanation:** This instruction contains no separate input materials. Therefore, the entire text is considered the core instruction.

**Example 2**

**Given Instruction:**

"Design a syllabus for the given course. Students should be given a list of the chapters with brief explanations of each chapter's purpose. \n\nProgramming for Everybody (Getting Started with Python)"

**Decomposition:**

- **Input Materials:** "Programming for Everybody (Getting Started with Python)"

- **Core Instruction:** "Design a syllabus for the given course. Students should be given a list of the chapters with brief explanations of each chapter's purpose."

**Explanation:** The course title serves as input material, separated from the core instruction by two newline characters.

{{...}}

Table 5: Human Annotation Guidelines for Instruction Decomposition Task

| Category | Dimension | Specification | Examples |
|---|---|---|---|
| Content | Include Content | Specify themes, concepts, or information that should be present. | Discuss the scientific research related to the 5-second rule. |
| | Exclude Content | Specify themes, concepts, or information that should be omitted. | Avoid mentioning urban legends not related to the 5-second rule. |
| | Narrow Scope | Define specific boundaries for the topic. | Focus solely on bacterial transfer rates on different surfaces. |
| | Expand Scope | Broaden the scope of discussion. | Explore cultural perceptions of food safety beyond the 5-second rule. |
| | Connect Elements | Establish relationships between concepts or ideas. | Link the concept of the 5-second rule to food safety regulations |
| | Compare Elements | Analyze similarities and differences between concepts or ideas. | Analyze the differences in contamination risk between wet and dry foods. |
| | Analyze Elements | Break down and examine parts or aspects of a topic. | Examine the role of surface type in bacterial transfer. |
| | Provide Examples | Suggest real-world or theoretical instances. | Include studies that have tested the 5-second rule in laboratory settings. |
| Style | Writing Style | Define the style or genre to write in or express. | Use an academic writing style. |
| | Establish Tone | Set the tone or attitude of the response. | Maintain a neutral tone throughout. |
| | Express Emotion | Convey emotions in the response. | Convey a sense of skepticism regarding the validity of the rule. |
| | Use Rhetoric | Employ rhetorical techniques. | Employ metaphors to explain bacterial movement. |
| Situation | Create Situation | Establish a specific context or setting. | Set the discussion in a high school science class. |
| | Assume Role | Assign a particular role or perspective. | Assume the perspective of a parent explaining to their child. |
| | Specify Timeline | Specify temporal context or chronological parameters. | Discuss the 5-second rule in the context of the 1990s. |
| | Establish Purpose | Clarify the reason or goal why the instruction is given. | The goal is to debunk common food safety myths. |
| Format | Output Format | Define the response format or pattern. | Present the information in a FAQ format. |
| | Create Hierarchy | Establish a hierarchical order for presenting content. | Organize information by level of scientific evidence. |
| | Structure Layout | Determine how information is organized. | Divide the content into sections with clear headings. |
| | Apply Template | Use a specific template or structure. | Use a case study format to explore the topic. |
| Numerical | Limit Length | Restrict the response length in terms of paragraphs or sentences. | Restrict to two sentences per section. |
| | Specify Quantities | Define the required number of items. | Include exactly three scientific studies in the discussion. |
| | Define Ranges | Set numerical ranges for elements. | Set a timeframe of 5-10 years for study references. |

Table 6: XIFBench's Constraint Taxonomy and Examples

| **Constraint Brainstorming Task** |
| --- |
| Your task is to brainstorm potential constraints for the "given instruction". The resulting "brainstormed constraints" will be sampled and combined with the "given instruction" later.
The goal is to produce a diverse range of constraints that naturally increase the difficulty of the "given instruction", without relying on language-specific features that don't translate well. |

| **Constraint Definitions and Categories**
{{CONSTRAINT_TAXONOMY}} |
| --- |

**Brainstorming Requirements**
- Generate {brainstormed_constraint_count} diverse constraints for each category.
- Make each constraint focus on a single, specific intention or requirement.
- Ensure all constraints are independent and combinable with each other.
- Never provide specific examples or solutions in the constraints.
- Avoid using language-specific constraints which may not apply universally across languages. For example:
  - When using rhetoric, avoid alliteration, puns, wordplay, and similar devices.
  - When specifying output format, avoid uppercase letters, punctuation marks, and similar elements.
  - When setting length parameters, avoid word counts, character counts, and similar restrictions.

**Output Format**
Provide the "brainstormed constraints" only using bullet points for each constraint. For example:
### Content Constraints
- <constraint direction>: <constraint specification>
- Include Content: xxx

**Task Inputs & Outputs**
Here's the "given instruction". Let's brainstorm potential constraints!
### Given Instruction
{given_instruction}
### Brainstormed Constraints

Table 7: Prompt for Constraint Brainstorming Step

| **Constraint Combination Task** |
| --- |
| Your task is to combine the "given instruction" with all "brainstormed constraints". The resulting "combined instruction" serves as a request or query for Large Language Models (LLMs).
The goal is to enhance the complexity of the "given instruction" by integrating the provided "brainstormed constraints". |

**Combination Requirements**
1. Integrate all "brainstormed constraints" reasonably into the "given instruction" without adding or omitting any constraints.
2. Avoid introducing any new details or requirements that are not present in the original "brainstormed constraints".
3. Express each constraint from the "brainstormed constraints" while maintaining its original category's intent.
4. Preserve the essential information from the "given instruction".
5. Craft a natural and flowing "combined instruction" by:
   - Separate meta-constraint from each main constraint and combine them when appropriate for clarity.
   - Arrange constraints order to enhance readability, such as from general to specific.
   - Using guiding words (like "please", "ensure") and connecting words (like "while", "also") to enhance flow.

**Output Format**
Provide the "combined instruction" only. For example:
```
<combined instruction>
```

**Examples**
{{...}}
### Given Instruction
Write a code block in Markdown containing an example of a code block in Markdown. Don't forget those quadruple backticks.
### Brainstormed Constraints
- situation constraint - establish purpose: aim to provide a guide for using markdown in collaborative document creation.
- format constraint - specify output format/pattern: use bold text to highlight markdown syntax within the explanation.
- content constraint - compare elements: discuss differences between inline code syntax and block code syntax in markdown.
### Combined Instruction
As a guide for collaborative Markdown documentation, please create a code block in Markdown that contains an example of a code block in Markdown. Don't forget those quadruple backticks. Also, please explain the differences between inline code syntax and block code syntax in Markdown, using bold text to highlight all Markdown syntax elements in your explanation.
{{...}}

**Task Inputs & Outputs**
Here are the "given instruction" and the "brainstormed constraints". Let's combine them and write a new "combined instruction"!
## Given Instruction
{given_instruction}
## Brainstormed Constraints
{brainstormed_constraints}
## Combined Instruction

Table 8: Prompt for Constraint Combination Step

**Requirement Decomposition Task**

Your task is to decompose the "given instruction" into "evaluation requirements".
The resulting "evaluation requirements" will be used to comprehensively and granularly assess whether a response adheres to the instruction.

**Decomposition Rules**
- Extract requirements only from the explicit wording of the instruction.
- Each requirement must correspond to a specific part of the instruction.
- Do not infer or add requirements that are not directly stated in the instruction.
- Ensure each requirement is atomic, focusing on a single, indivisible aspect of the instruction.
- Format each requirement as a YES/NO question that can be used to verify if the requirement has been met.

**Output Format**
Provide the "evaluation requirements" only using numbered lists like the following format:
1. `<evaluation requirement 1>`
2. ...

**Examples**
{{...}}
### Given Instruction
Please explain Fermat's Last Theorem within the context of an interview with a renowned mathematician, utilizing a FAQ template to address various aspects. Ensure the explanation is concise, limited to no more than three sentences.
### Evaluation Requirements
1. Does the response explain Fermat's Last Theorem?
2. Is the explanation presented within the context of an interview with a renowned mathematician?
3. Does the response utilize a FAQ template to address various aspects?
4. Is the explanation concise?
5. Is the explanation limited to no more than three sentences?
{{...}}

**Task Inputs & Outputs**
Here's the "given instruction". Let's decompose evaluation requirements!
## Given Instruction
{given_instruction}
## Evaluation Requirements

Table 10: Prompt for Requirement Decomposition Step

**Requirement Categorization Task**

Your task is to categorize each item in the "evaluation requirements" into the most appropriate "requirement category" and "requirement direction".

The "evaluation requirements" are derived from the "given instruction," which provides the contextual background for each requirement.

**Requirement Definitions and Categories**

{{CONSTRAINT_TAXONOMY}}

**Categorization Guidelines**

Requirements should be categorized through a three-step evaluation process with category prioritization:

1. First, trace back each requirement to its original intent in the "given instruction", rather than focusing on how it's phrased in the evaluation.

2. Second, evaluate if the requirement fits Numerical or Situation categories or not.

3. Third, if the requirement doesn't fit Numerical or Situation categories, then consider Format, Style, or the lastly Content categories.

**Output Format**

Provide the "requirement categories" in a numbered list. For example:

1. <requirement category of 1st requirement> - <requirement direction of 1st requirement>

2. <requirement category of 2nd requirement> - <requirement direction of 2nd requirement>

**Examples**

{{...}}

### Given Instruction

Please explain Fermat's Last Theorem within the context of an interview with a renowned mathematician, utilizing a FAQ template to address various aspects. Ensure the explanation is concise, limited to no more than three sentences.

### Evaluation Requirements

1. Does the response explain Fermat's Last Theorem?

2. Is the explanation presented within the context of an interview with a renowned mathematician?

3. Does the response utilize a FAQ template to address various aspects?

4. Is the explanation concise?

5. Is the explanation limited to no more than three sentences?

### Requirements Categories

1. Content Requirements - Include Content

2. Situation Requirements - Create Situation/Environment

3. Format Requirements - Apply Template

4. Style Requirements - Specify Writing Style

5. Numerical Requirements - Limit Length Parameters

{{...}}

**Task Inputs & Outputs**

Here's the "given instruction" and "evaluation requirements". Let's categorize them properly!

## Given Instruction

{given_instruction}

## Evaluation Requirements

{evaluation_requirements}

## Requirements Categories

Table 11: Prompt for Requirement Categorization Step

**Requirement Assessment Task**

**Task Introduction**
Your task is to assess the evaluation requirements of each instruction against specific rules given in the criteria section.
We will use these requirements as criteria to assess instruction-following capabilities of LLMs. While we will translate instructions into `Chinese`, `Russian`, `Arabic`, `Hindi`, and `Swahili`, evaluation requirements remain in English to ensure comparability.

**Input & Output Explanation**
{{INPUT_OUTPUT_EXPLANANTION}}

**Validation Criteria**
Please check each instruction against the following criteria step-by-step:

**Rule 1: Are the evaluation requirements explicitly stated in the instruction? [YES/NO]**
**Definition:** Evaluation requirements should be explicitly stated in the instruction, without any need for inference or assumption. The requirements should be directly derived from the content of the instruction to ensure that the evaluation criteria are based on the user's explicit intentions.
**Positive Example:**
   `instruction`: "What color goes best with teal?"
   `evaluation_requirements`: "1. content requirement - include content: Does the response identify a color going well with teal?"
   **Accepted Reason:** The requirement is explicitly present in the instruction, asking for a color that pairs well with teal.
**Negative Example:**
   `instruction`: "What color goes best with teal?"
   `evaluation_requirements`: "2. content requirement - analyze components: Does the response explain why the suggested color goes well with teal?"
   **Rejected Reason:** The requirement is not explicitly present in the instruction, as it asks for an explanation rather than a color pairing.

**Rule 2: Are the evaluation requirements complete? [YES/NO]**
**Definition:** Evaluation requirements should encompass all explicitly stated elements from the instruction. Each component directly mentioned in the instruction should have a corresponding evaluation requirement. This ensures that the evaluation criteria cover all explicit instructions.
**Positive Example:**
   `instruction`: "As a journalist covering border news, could you provide information on whether the US border is open to Canada?"
   `evaluation_requirements`:
     - "1. situation requirement - assume role: Does the response adopt the perspective of a journalist covering border news?"
     - "2. content requirement - include content: Does the response provide information on whether the US border is open to Canada?"
   **Accepted Reason:** The requirements are complete, covering all explicitly stated elements from the instruction.
**Negative Example:**
   `instruction`: "As a journalist covering border news, could you provide information on whether the US border is open to Canada?"
   `evaluation_requirements`:
     - "1. content requirement - include content: Does the response provide information on whether the US border is open to Canada?"
   **Rejected Reason:** The requirements are incomplete, missing the situation requirement to adopt the journalist's perspective.

**Rule 3: Are the evaluation requirements atomic? [YES/NO]**
**Definition:** Evaluation requirements should be atomic, focusing on a single, distinct aspect of the instruction. This means each requirement should be decomposed to the finest granularity where it can still be evaluated independently, without creating dependencies between requirements during evaluation.
**Positive Example:**
   `instruction`: "Create a short, concise summary of the paper based on its abstract."
   `evaluation_requirements`:
     - "1. content requirement - include content: Does the response provide a summary of the paper?"
     - "2. situation requirement - establish purpose: Is the summary based on the paper's abstract?"
     - "3. style requirement - specify writing style: Is the summary short?"
     - "4. style requirement - specify writing style: Is the summary concise?"
   **Accepted Reason:** The requirements are atomic because each one focuses on a distinct aspect of the instruction.

**Rule 4: Are the evaluation requirements properly categorized and directed? [YES/NO]**
**Definition:** Evaluation requirements should be properly categorized and directed based on their primary function within the instruction, tracing back to the original intent rather than the phrasing. These categories will be used to analyze the instruction-following capabilities of LLMs from different perspectives. See {{CONSTRAINT_TAXONOMY}} for detailed definitions and categories.
**Categorization Priority:** Follow this order when categorizing requirements:
   1. Prioritize Numerical and Situation categories first, as they typically have the most explicit indicators.
   2. If the requirement does not fit into Numerical or Situation categories, then evaluate Format, Style, and finally Content categories.
**Relation between Categorization and Direction:**
   1. First, evaluate the requirement's category based on its primary function.
   2. Next, evaluate the direction based on the instruction's intent, ensuring the requirement aligns with the instruction's purpose.
   3. Typically, if requirements are miscategorized, they will also be improperly directed.

**Task Workflow**
1. Read the `instruction` and `evaluation_requirements` fields.
2. Evaluate each `evaluation_requirements` against the validation criteria.
3. If feeling uncertain, you can (recommended from most to least encouraged):
   - Use web search or external LLMs to understand the instruction and requirements better.
   - Test with an LLM by:
     a. Input the instruction to get a response.
     b. Use the evaluation requirements to assess the response.
     c. Compare if your assessment aligns with the requirements' intended evaluation.
4. Fill in the `human_annotation_tags` fields based on your assessment.
5. Add any additional comments or notes in the `notes` field if necessary.

Table 12: Human Annotation Guideline for Requirement Assessment Task

**Translation Validation Task**

Your task is to evaluate and score the "translated instruction" in {translated_language} to determine if it maintains the essential requirements of the "original instruction," as detailed in the "evaluation requirements." Use both the "translated instruction" and the "back-translated instruction" to assess accuracy.

Your goal is to ensure all essential requirements from the original instruction are preserved in the translation.

**Input Definitions**

**Original Instruction**: The original instruction that needs to be translated.
**Translated Instruction**:
  - The machine-translated version of the "original instruction" in {translated_language}.
  - It is the instruction needing to be evaluated for quality.
**Back-translated Instruction**:
  - The English machine-translated version of the "translated instruction".
  - It serves as an additional tool to verify the translation's accuracy.
**Evaluation Requirements**:
  - **Definition**: YES/NO questions designed to assess whether a response follows the instruction. Each question corresponds to an explicit and atomic requirement from the original instruction.
  - **Purpose**: During evaluation, each question is used to locate the corresponding portions in the translation that need to be evaluated.
  - **Metadata**: Information such as "content requirement - include content" indicates the category and direction of the requirement, aiding in identifying the portion to be evaluated.
  - **Implicit Requirements**: Occasionally, there may be implicit requirements not explicitly stated in the instruction. These should be identified and treated as exceptions (detailed below) to ensure they do not affect the overall validation.

**Special Case: Implicit Requirements**

Implicit requirements are requirements that are **not directly present or reflected in the "original instruction"** but may have been inferred or included in the "evaluation requirements" due to automated processing.

Do not evaluate the "translated instruction" for implicit requirements, as they do not exist in the original instruction and cannot be preserved in the translation. In this case, implicit requirements should be treated as exceptions and handled as follows:
1. Replace all portions (Original, Translated, and Back-translated) with the placeholder "[IMPLICIT_REQUIREMENT]".
2. Provide a concise observation explaining why the requirement is implicit.
3. Assign a score of "0/5" to indicate that the requirement is not applicable to the translation evaluation.

**Scoring Scale**

The scoring scale ranges from 1 to 5, with 3/5 as the acceptance threshold for translation quality.
- **5/5 (Perfect Preservation)**: Core requirements are preserved with complete semantic accuracy.
- **4/5 (Strong Preservation)**: Core requirements are maintained with very minor variations.
- **3/5 (Acceptable Preservation)**: Core requirements are maintained with some variations.
- **2/5 (Weak Preservation)**: Core requirements are partially distorted or unclear.
- **1/5 (Failed Preservation)**: Core requirements are significantly distorted or completely lost.
Remember to use a score of 0/5 as a special placeholder for implicit requirements, indicating their inapplicability.

**Evaluation Guidelines**

For each "evaluation requirement," follow these steps to assess the quality of the translation:
1. **Portion Identification**:
  - Extract the minimal relevant portion from "original instruction", "translated instruction", and "back-translated instruction" using the "evaluation requirements" as a guide.
  - If the instruction combines multiple requirements, isolate only the portion relevant to the current requirement to ensure focusing.
  - For implicit requirements, replace all those 3 portions with the placeholder "[IMPLICIT_REQUIREMENT]".
2. **Observation**: Provide a concise observation focusing on:
  - How well the "translated portion" in "translated instruction" preserves the semantics of the "original portion" in "original instruction"?
  - Use the "back-translated portion" in "back-translated instruction" to verify the accuracy of the translation.
  - For implicit requirements, explain why the requirement is not applicable.
3. **Scoring**: Assign a score from 1 to 5 in the format "X/5" based on the scoring scale.
  - For implicit requirements, always assign a score of "0/5".

**Output Format**

Ensure your evaluation for each requirement is clearly structured as follows, without any additional information:
### Requirement {Number}: {Requirement Description}
#### Portions
- Original Portion: [Minimal relevant portion from "original instruction"]
- Translated Portion: [Minimal relevant portion from "translated instruction"]
- Back-translated Portion: [Minimal relevant portion from "back-translated instruction"]
#### Observation
{Provide a concise observation on semantic preservation and accuracy. For implicit requirements, explain why.}
#### Scoring
{Assign a score from 1 to 5 based on the scoring scale, formatted as "X/5". For implicit requirements, use "0/5".}

**Task Inputs & Outputs**

| | |
|---|---|
| ### Original Instruction | ### Back-translated Instruction |
| {original_instruction} | {back_translated_instruction} |
| ### Translated Instruction (in {translated_language}) | ### Evaluation Requirements |
| {translated_instruction} | {evaluation_requirements} |
| ### Your Validations | |

Table 14: Prompt for Translation Validation Step

**Instruction Following Evaluation Task**

Your task is to evaluate whether the "model response" follows the "English instruction" by checking if it satisfies all "evaluation requirements".

Your goal is to assess the instruction following quality of the "model response" using binary (YES/NO) decisions for each evaluation requirement.

**Input Definitions**

• **English Instruction:**
  - The English instruction given to a Large Language Model (LLM).
  - It also serves as the source for deriving evaluation requirements.
• **Model Response:**
  - The LLM's response to the English instruction.
  - It will be evaluated against each requirement.
• **Evaluation Requirements:**
  - The YES/NO questions designed to assess whether the response follows the instruction. Each question corresponds to an explicit and atomic requirement from the English instruction.
  - Additional metadata (e.g., "content requirement - include content") provides context about the type and focus of the requirement. Use this metadata to identify the specific aspect of the response that needs to be evaluated.

**Evaluation Rules**

For each evaluation requirement, provide a **strict** YES/NO decision based on the following principles:
• **YES:** Select "YES" **only if** the response **fully satisfies** the requirement **without any errors, omissions, ambiguities, or deviations**, even minor ones.
• **NO:** Select "NO" if the response **fails to satisfy the requirement, contains inaccuracies, omits relevant details, or introduces ambiguity**.

This strict evaluation means that even if the response is **mostly correct**, **partially correct**, or **correct under certain conditions**, it should still be evaluated as "NO".

**Output Format**

Provide a structured evaluation for each requirement in order based on the following format, without any additional information.
## Requirement {Number}: {Requirement Description}
### Observation
 [Provide a concise observation on how the "model response" satisfies the requirement.]
### Decision
 [Output your evaluation decision (YES/NO) for the requirement only.]

**Task Inputs & Outputs**

## English Instruction
{english_instruction}
## Model Response
{model_response}
## Evaluation Requirements
{evaluation_requirements}
## Your Evaluation

Table 15: Prompt for the Instruction Following Evaluation Task on English Instructions.

**Instruction Following Evaluation Task**

Your task is to evaluate whether the "model response" follows the "{translated_language} instruction" by checking if it satisfies all "evaluation requirements".

Your goal is to assess the instruction following quality of the "model response" using binary (YES/NO) decisions for each evaluation requirement.

**Input Definitions**

• **{translated_language} Instruction:**
  - The {translated_language} instruction given to a Large Language Model (LLM).
  - The directive part is translated from the English instruction, while other parts remain unchanged intentionally.
• **Model Response:**
  - The LLM's response to the {translated_language} instruction.
  - It will be evaluated against each requirement.
• **English Instruction:**
  - The original English instruction from which the {translated_language} instruction is translated. Unchanged parts are omitted for brevity.
  - It serves as the source for deriving evaluation requirements and helps in understanding the {translated_language} instruction.
• **Evaluation Requirements:**
  - The YES/NO questions designed to assess whether the response follows the instruction. Each question corresponds to an explicit and atomic requirement from the English instruction.
  - Additional metadata (e.g., "content requirement - include content") provides context about the type and focus of the requirement. Use this metadata to identify the specific aspect of the response that needs to be evaluated.

**Evaluation Rules**

For each evaluation requirement, provide a **strict** YES/NO decision based on the following principles:
• **YES:** Select "YES" **only if** the response **fully satisfies** the requirement **without any errors, omissions, ambiguities, or deviations**, even minor ones.
• **NO:** Select "NO" if the response **fails to satisfy the requirement, contains inaccuracies, omits relevant details, or introduces ambiguity**.

This strict evaluation means that even if the response is **mostly correct**, **partially correct**, or **correct under certain conditions**, it should still be evaluated as "NO".

**Output Format**

Provide a structured evaluation for each requirement in order based on the following format, without any additional information.
## Requirement {Number}: {Requirement Description}
### Observation
 [Provide a concise observation on how the "model response" satisfies the requirement.]
### Decision
 [Output your evaluation decision (YES/NO) for the requirement only.]

**Task Inputs & Outputs**

## {translated_language} Instruction
{translated_instruction}
## Model Response
{model_response}
## English Instruction
{english_instruction}
## Evaluation Requirements
{evaluation_requirements}
## Your Evaluation

Table 16: Prompt for the Instruction Following Evaluation Task on Translated Instructions.

**Instruction Following Evaluation Assessment**

**Task Introduction**

Your task is to assess the decisions made by a GPT-4o based evaluator and determine whether you agree with them.

This multilingual instruction-following benchmark covers six languages: English (en), Chinese (zh-cn), Russian (ru), Arabic (ar), Hindi (hi), and Swahili (sw). Your evaluations will help us **measure the agreement of GPT-4o based evaluators with humans across different languages**.

**Input & Output Explanation**

You will receive a **TXT file** and a **JSON file**, each containing the prompts, responses and GPT-4o's evaluations. Each item in both files shares the same unique ID.

{{INPUT_OUTPUT_EXPLANATION}}

**Output Explanation**

You need to review each decision made by the GPT-4o evaluator and indicate whether you agree with their assessment. Record your agreement in the `instruction_following_evaluation` section of the JSON file under the following fields:

- `do_agree`: Set to `true` if you agree with the evaluator's decision. Set `false` if you disagree.
- `note`: (Optional) Provide a **brief reason** if you **disagree** with the evaluator's decision.

**Important Notes:**

- The initial `true` or `false` values in `do_agree` are **placeholders** – they should **not influence your judgment**.
- While the `note` field is optional, it is **strongly recommended** when you disagree, as it helps us to analyze disagreement cases.

**Task Guidelines**

The satisfaction of each requirement in `decision` should be evaluated with a **strict** binary approach (`true` or `false`). Below are the principles of judgment for both the GPT-4o evaluator and you:

- `true`: Select `true` **only if** the response **fully satisfies** the requirement **without any errors, omissions, ambiguities, or deviations**, no matter how minor.
- `false`: Select `false` if the response **fails to satisfy the requirement, contains inaccuracies, omits relevant details, or introduces ambiguity**.

This strict evaluation means that even if the response is **mostly correct**, **partially correct**, or **correct under certain conditions**, it should still be evaluated as `false`.

**Task Examples**

The following examples illustrate **how strict GPT-4o and you should be** for assessing instruction-following quality. These are **not exhaustive** but should guide your judgment.

- *content requirement - connect elements: Does the response connect the meaning of life to human happiness?*
  Select `false` if the response **relates the meaning of life to human happiness improperly** or **connects elements ambiguously**.
- *style requirement - specify writing style: Is the response written in an academic style?*
  Select `false` if the response **contains informal language** or **lacks academic rigor**.
- *situation requirement - assume role: Does the response adopt the perspective of a historian?*
  Select `false` if the response **does not clearly adopt a historian's perspective** or **rarely implies a historian's viewpoint**.
- *situation requirement - establish purpose: Does the response aim to persuade the reader?*
  Select `false` if the response **fails to persuade the reader** or **persuades the reader ambiguously**.
- *format requirement - specify output format/pattern: Is the response formatted as a checklist?*
  Select `false` if the response **lacks clear checkboxes, bullet points or numbering**.
- *numerical requirement - limit length parameters: Does the response have exactly five sentences?*
  Select `false` if the response **has more or fewer than five sentences**.

**Task Workflow**

1. Read the `original_instruction` in the JSON file to understand the task requirements.
2. For each requirement in `instruction_following_evaluation`, follow these steps:
   - Read the `requirement` and `observation` to understand the context.
   - Read the [PROMPT] and [RESPONSE] sections in the TXT file to examine the actual prompt and response.
   - Use **Google Translate** or other tools if needed to understand the [PROMPT] and [RESPONSE] content.
   - Determine if the [RESPONSE] fully meets the requirement.
   - The `observation` can help navigate the response, but **do not rely on it as evidence** for agreement or disagreement, as it may not always accurately reflect the response.
3. Update the `do_agree` field based on your assessment for each requirement.
4. If you disagree with the evaluator's decision, provide a **brief reason** in the `note` field (if possible).

Table 17: Human Annotation Guideline for Instruction Following Evaluation Assessment Task

