# OpenReview forum: "XIFBench: Evaluating Large Language Models on Multilingual Instruction Following"
_NeurIPS.cc/2025/Datasets_and_Benchmarks_Track — NeurIPS 2025 Datasets and Benchmarks Track poster_

### Official Review · Reviewer_Y8vR · 2025-06-24

**Rating:** 4
**Confidence:** 4

**Summary:**

The paper describes a new benchmark called XIFBench, which is aimed to measure the capabilities of SOTA LLMs in following diverse fine-grained instruction following across multiple languages. Specifically, XIFBench is composed of 558 instructions synthetically generated through translations using an automated framework (Constraint Augmentation, Requirement Structuring, Multilingual Expansion) from already existing benchmark datasets including ALPACA, and WizardLM, LIMA. The authors motivate the need for said benchmark as current benchmarks for constraint based following mainly focus on English or Chinese, and most do not have a wide coverage of medium and low-resource languages whereas XIFBench covers six languages (English, Chinese, Russian, Arabic, Hindi, and Swahili).  Ablations conducted by the authors in line with multilinguality and effectiveness of the benchmark include investigating effects of categories and instruction complexity for crosslingual performance of models as well as cultural specificity. Main results of the paper show that commercial models outperform open source models for the benchmark, specifically Gemini 2.0 Flash and similar results are observed for this model on cross-lingual constraint coherence  and consistency in performance across increasing instruction levels.

While the benchmark may serve its purpose as secondary resource for evaluating capabilities of LLMs especially multilingual ones (which the authors did not include in their experiments), there are several methodological and reliability issues in the paper that prevents me from confidently favoring acceptance for the work.

**Dataset Code Accessibility:**

Yes

**Dataset Code Comments:**

The dataset is available in a public link.

**Ethical Considerations:**

No, there are no or only very minor ethics concerns

**Final Justification:**

I increased my score from 3 to 4, given that the authors have provided thorough additional experiments that addressed most of my concerns. Certain inherent limitations of the XIFBench such as being machine translated but post-checked by native speakers should be clearly discussed in the paper. Additionally, my recommended additional experiments on performance of multilingual models and carry-over effect might help the authors strengthen the discussion section of the paper.

**Limitations Weaknesses:**

One of the main concerns I have with the benchmark is the lack of clear discussion on its language validity and practical usage. The benchmark has been adopted/translated from existing benchmarks (ALPACA, Lima, WizardLM) which are publicly available and have already been saturated or included in training data of most SOTA LLMs, hence explains why the performance of the best models are in the 90s. This should be clearly discussed in the main section of the paper. This is the first piece of information I was trying to find and it’s not clearly discussed in the paper and pushed all the way back in the Appendix. Moreover, in the Appendix, it is mentioned that the validation procedure only made use of a sample of instances per language and only one native speaker per language, which further reduces the reliability of this benchmark.

Given the main concern with LLM benchmarks these days being saturated and the reliability of their evaluations on LLM performance, the onus is on the authors to clearly discuss how their proposed benchmark (XIFBench) will maintain its usabilit and lifetime in the next few months. Are there any plans on continuously building or expanding to other languages? Will a subset of the benchmark be withheld to protect from memorization? Aside from this, I believe the paper should come with an extensive disclaimer in the Limitations section that the benchmark is mainly LLM-translated from existing benchmarks and another LLM is used to evaluate the performance. Even if it’s an upgraded version, they still belong to the same model family and potentially use the same architecture, post-training data, etc might explain why GPT-4o obtains favorable performance.

The paper needs to investigate a “carry-over effect” where results from the base benchmarks (LIMA, Alpaca, WizardLM) might potentially be the similar to what LLMs evaluated with XIF might obtain it is not an original benchmark dataset measuring novel constraints. This should be done language-wise as well. I think the best way to do this form of analysis is to group the types of tests per source benchmark and see its corresponding translations per language have some form of similarities between them. Moreover, another angle that the authors can also explore possible benchmark score correlations where if model X performs well existing multilingual benchmarks like M-IFEval or MultiIF, it might also perform well on XIF. Results from these experiments might help dispel future readers’ assumptions that XIF is yet another multilingual constraint-based dataset that is closely similar to M-IFEval or MultiIF and/or translated from existing ones using GPT-4o.

The paper describes and introduces a multilingual benchmark, and yet, no open multilingual models were evaluated. I believe this is also a major weakness of the paper as this defeats the purpose of the a multilingual benchmark which researchers interested in the field will also have multilingual models as one of their resources. There are an abundance of open multilingual models that should be added in the evaluation experiments including Aya101, Gemma3, EuroLLM, Aya23, BLOOM, etc.

**Strengths Contributions:**

The paper is well-crafted, and I recognize the evidence from the conceptualization to the discussion of results that the authors really put effort into constructing this multilingual benchmark. While the benchmark is not originally human-authored, it may serve a purpose for some form of evaluation where researchers are looking for versions of constraint-based testing in different languages.

---

> ### Author Rebuttal · Authors · 2025-07-31
>
> We sincerely appreciate your comprehensive, perceptive, and rigorous feedback. Please find our responses to your valuable suggestions below.
>
> ---
>
> **LW1.1**: We appreciate the opportunity to clarify and fully agree that this discussion deserves more visibility in the main paper.
>
> XIFBench reuses instructions from existing benchmarks, but its Hard Set is newly designed via a systematic constraint augmentation process. This results in new English instructions with diverse constraint types and controllable difficulty levels, followed by human validation. Our construction strategy aligns with recent benchmarks. We summarize the comparison below:
>
> | Benchmark | Source Data | Instruction Construction Method |
> |---------------|--------------------------------------------------|----------------------------------------------|
> | FollowBench | Diverse datasets and benchmarks | Add constraints & rewrite via human & LLM |
> | InfoBench | `Self-Instruct` for its Easy Set | Expert crafts its Hard Set |
> | IFEval | Unknown | Add constraints & rewrite via LLM |
> | CFBench | Real-world data | Add constraints & rewrite via LLM |
> | ComplexBench | `FollowBench` + `InfoBench` + `IFEval` | Add constraints & rewrite via human |
> | Multi-IF | `IFEval` | Add constraints & rewrite via LLM |
> | M-IFEval | `IFEval` | Human-modified language-specific instructions |
>
> We agree that some high-resource languages and closed-source models achieve strong RFRs (up to 93.6%). However, XIFBench still highlights large gaps across mid-/low-resource languages and model types—e.g., open-source models range from 89.2% to 10.0% RFR—underscoring its value in multilingual evaluation. Also, the overall IFR (e.g., 78.1% for GPT-4o) remains far from perfect, suggesting headroom remains.
>
> We will bring these discussions to the main paper in the revised version.
>
> ---
>
> **LW1.2**: We have expanded our human auditing process as follows:
>
> - We increased the human audit coverage from 10% (10 Easy + 50 Hard English instructions, with their translations) to 20% (18 Easy + 90 Hard)
> - Instead of one annotator per language, each language is now evaluated by two native-speaking annotators who score independently.
> - We report the lower of the two scores for each translation to conservatively estimate translation quality.
>
> The updated translation quality scores (on a 1–5 scale) are summarized in the table below. Over 97.7% of translations received a score ≥ 3, indicating good quality. These results will be included in Appendix C.4 of the revised version.
>
> | Score | Zh-cn | Ru | Ar | Hi | Sw |
> |-------|-------|------|------|------|------|
> | 5 | 81.5 | 79.5 | 83.0 | 76.8 | 79.3 |
> | 4 | 13.3 | 15.9 | 12.0 | 17.8 | 15.9 |
> | 3 | 4.2 | 3.3 | 2.7 | 4.3 | 3.7 |
> | 2 | 1.0 | 0.8 | 2.3 | 0.8 | 1.0 |
> | 1 | 0.0 | 0.4 | 0.0 | 0.2 | 0.0 |
>
> ---
>
> **LW2.1**: We fully agree that long-term usability is essential for a benchmark. To this end, we are working on extending XIFBench towards a live benchmark by continuously sampling challenging instructions and constraints from real-world dialogs (e.g., WildChat). Our pipeline is designed to be scalable, enabling continuous updates.
>
> To futher examine multilingual generalization, we stratified and sampled one-third of XIFBench and translated it into: French, Spanish, Turkish, Vietnamese, Urdu, and Tamil. Due to space constraints, we only report RFR results below. These findings indicate that XIFBench can reveal meaningful performance gaps in multilingual settings.
>
> **RFR**:
>
> | Model | Fr | Es | Tr | Vi | Ur | Ta | Avg. |
> |----------------------|------|------|------|------|------|------|------|
> | Gemini-2.0-Flash | 90.9 | 91.7 | 89.0 | 91.4 | 87.5 | 80.3 | 88.5 |
> | Qwen2.5-72B-Instruct | 86.1 | 87.7 | 89.0 | 86.9 | 76.8 | 53.4 | 79.9 |
> | GLM-4-9B-Chat | 81.4 | 83.0 | 80.2 | 80.6 | 58.6 | 55.9 | 73.3 |
>
> Additionally, we are exploring the implementation of a leaderboard or shared-task to facilitate continuous evaluation under controlled conditions.
>
> ---
>
> **LW2.2**: We will add a clear disclaimer in the Limitations section in our revised version.
>
> ---
>
> **LW2.3**: We agree that model family similarities (e.g., architecture or post-training data) may contribute to GPT-4o's performance. To cross-check our findings, we additionally applied **Gemini-2.5-Flash**—a model from a different family (human agreement: 89.2%)—to re-evaluate GPT-4o. The results are consistent with those reported in the paper, as shown below:
>
> **RFR**:
>
> | Settings                  | En   | Zh-cn | Ru   | Ar   | Hi   | Sw   | Avg. |
> |---------------------------|------|-------|------|------|------|------|------|
> | Evaluated as in Paper     | 93.6 | 92.5  | 92.7 | 90.8 | 92.8 | 90.8 | 92.2 |
> | Evaluated by Gemini-2.5-Flash | 93.3 | 93.1  | 92.2 | 90.4 | 91.6 | 90.7 | 91.9 |
>
> **IFR**:
>
> | Settings                  | En   | Zh-cn | Ru   | Ar   | Hi   | Sw   | Avg. |
> |---------------------------|------|-------|------|------|------|------|------|
> | Evaluated as in Paper     | 76.9 | 73.3  | 74.2 | 69.2 | 73.8 | 65.6 | 72.2 |
> | Evaluated by Gemini-2.5-Flash | 75.3 | 73.6  | 71.1 | 68.8 | 70.8 | 66.1 | 71.0 |
>
> ---
>
> **LW3.1**: To investigate the potential *carry-over effect*, we analyzed correlations between the Easy Set (sampled from existing benchmarks) and the Hard Set (newly constructed instructions), grouped by source (AlpacaEval, WizardLM, LIMA) and across six languages. We report Pearson correlations under RFR and IFR metrics.
>
> **RFR Pearson Correlation**
>
> |           | En   | Zh-cn | Ru   | Ar   | Hi   | Sw   |
> |-----------|------|-------|------|------|------|------|
> | AlpacaEval | 0.69 | 0.91  | 0.72 | 0.97 | 0.95 | 0.98 |
> | WizardLM   | 0.49 | 0.76  | 0.73 | 0.84 | 0.73 | 0.97 |
> | LIMA       | 0.70 | 0.93  | 0.73 | 0.86 | 0.58 | 0.99 |
> | **Avg.**   | 0.63 | 0.87  | 0.73 | 0.89 | 0.75 | 0.98 |
>
> **IFR Pearson Correlation**
>
> |           | En   | Zh-cn | Ru   | Ar   | Hi   | Sw   |
> |-----------|------|-------|------|------|------|------|
> | AlpacaEval | 0.88 | 0.87  | 0.81 | 0.92 | 0.95 | 0.97 |
> | WizardLM   | 0.68 | 0.66  | 0.78 | 0.93 | 0.58 | 0.89 |
> | LIMA       | 0.76 | 0.87  | 0.85 | 0.75 | 0.64 | 0.95 |
> | **Avg.**   | 0.77 | 0.80  | 0.81 | 0.87 | 0.72 | 0.94 |
>
> These results show moderate to strong correlations, especially in high-resource languages, suggesting the Hard Set is related but not redundant.
>
> In low-resource languages like Swahili, higher correlations may reflect limited instruction diversity. However, model-level differences remain: e.g., for LIMA-Swahili, Qwen-2.5-72B drops from RFR 40.0→31.0, IFR 10.5→2.1 between Easy and Hard Sets. This suggests the Hard Set still introduces meaningful challenges. We believe these high correlations should be interpreted with cautious, possibly reflecting reduced headroom for differentiation in low-resource settings rather than redundancy in instruction design.
>
> ---
>
> **LW3.2**: Due to time constraints, we conducted a preliminary analysis using models evaluated on all three datasets (XIFBench, Multi-IF, and M-IFEval). The table below shows the average full-language scores for XIFBench (RFR & IFR), alongside the average scores reported in Table 1 of Multi-IF and M-IFEval:
>
> | Model | XIFBench | Multi-IF | M-IFEval |
> |--------------------|----------|----------|----------|
> | GPT-4o | 82.2 | 84.3 | 82.7 |
> | Claude-3.5-Sonnet | 66.0 | 81.7 | 84.2 |
> | Qwen-2.5-72B | 66.0 | 83.7 | - |
> | Llama-3.1-70B | 65.7 | 82.6 | - |
>
> We observe a Pearson correlation of 0.71 between XIFBench and Multi-IF, suggesting moderate alignment in model ranking. Notably, Claude 3.5 performs well on M-IFEval but relatively lower on XIFBench, likely due to its weaker instruction-following performance in mid- and low-resource languages, which are more emphasized in XIFBench.
>
> We will extend this comparison to more models and metrics in our revised version.
>
> ---
>
> **LW4**: We appreciate your suggestion to include multilingual models. This indeed complements our current focus on general-purpose LLMs.
>
> Following your advice, we evaluated several multilingual models of comparable size:
>
> - **CohereLabs/aya-101** (13B, 101 languages)
> - **utter-project/EuroLLM-9B-Instruct** (9B, 35 languages)
> - **CohereLabs/aya-23-8B** (8B, 23 languages)
> - **bigscience/bloomz-7b1** (7B, 46 languages)
>
> We encountered unresolved inference issues (cache length mismatching) with **google/Gemma-3-12B-it** during evaluation, so we excluded it for now.
>
> We followed the same inference and evaluation setup as in Section 4.1. For models like aya-101 and bloomz-7b1 that lack user-assistant prompting templates, we used customized system prompts and simulated dialogue formatting to mitigate instruction continuation issues. The results are summarized below:
>
> **RFR:**
>
> | Model | en | zh-cn | ru | ar | hi | sw | avg |
> |-------|----|-------|----|----|----|----|-----|
> | aya-23-8B | 79.3 | 78.3 | 78.7 | 77.1 | 72.6 | 4.7 | 65.1 |
> | EuroLLM-9B | 73.2 | 74.5 | 70.4 | 62.3 | 58.4 | 8.8 | 57.9 |
> | aya-101 | 16.6 | 13.0 | 15.7 | 14.8 | 13.3 | 15.1 | 14.8 |
> | bloomz-7b1 | 8.1 | 6.3 | 4.2 | 7.1 | 6.4 | 3.3 | 5.9 |
>
> **IFR:**
>
> | Model | en | zh-cn | ru | ar | hi | sw | avg |
> |-------|----|-------|----|----|----|----|-----|
> | aya-23-8B | 41.8 | 41.0 | 43.5 | 39.5 | 31.7 | 1.5 | 33.2 |
> | EuroLLM-9B | 35.7 | 38.4 | 32.4 | 23.8 | 21.2 | 1.8 | 25.6 |
> | aya-101 | 3.4 | 3.1 | 3.1 | 2.9 | 2.5 | 3.4 | 3.1 |
> | bloomz-7b1 | 2.0 | 1.8 | 1.6 | 2.3 | 2.3 | 0.9 | 1.8 |
>
> We observed the following:
>
> - Aya-23-8B consistently outperforms EuroLLM-9B-Instruct on all languages except Swahili. This may be due to its more balanced multilingual training data (CohereLabs/aya_collection), whereas EuroLLM-9B appears more English and Western-European centric.
> - Both aya-101 and bloomz-7b1 lack user-assistant instruction tuning, which may contribute to their poor performance on instruction-following.
>
> We will incorporate this discussion into the revised version.

---

> > ### Comment · Reviewer_Y8vR · 2025-08-03
> > **Acknowledgment of authors' response and additional experiments**
> >
> > This is to formally acknowledge that I have read the authors' responses as well as my co-reviewers' feedback. I very much appreciate the depth that the authors went through with the experiments in line with responding to my feedback.
> >
> > Regarding the benchmark lifeline, what I mean is that the paper should have a specific discussion (maybe a short part of the introduction and in the conclusion) on how this benchmark's data can be protected from saturation. I suggest that the authors explore appropriate licensing for the benchmark dataset (e.g., CC-BY-NC) so that it is protected from commercial model development use. Moreover, privately withholding a portion of the test set should also be in place.
> >
> > Regarding experiments on multilingual models, this should have been done in the first place. Nonetheless, it is surprising to see that heavily multilingual models (Aya101, Bloomz) are less performant than selectively multilingual models (Aya-23, EuroLLM). I strongly suggest the authors to center their main discussions around these models as it exposes their limitations + well-aligned with the multilinguality of the benchmark.
> >
> > Regarding the high correlation scores as a potential carry-over effect, this might also be an insightful piece that the authors can discuss in their paper. Obviously, there will be other benchmarks that follow the same methodology where portions of the dataset are adopted from existing benchmarks (e.g., GLUE-X, MixEval, BenchHub) . But setting the tone and discussing carry-over effects might also help readers understand the limitations of this particular process in the LLM benchmarking domain. I don't particularly hold this as a weakness (the high correlation score as a carry-over effect) but an opportunity to address and improve practices in benchmark building. I hope this helps the authors in writing about this insight in the paper.
> >
> > I'm increasing my score for the paper in line with authors' response.

---

> > ### Author Response · Authors · 2025-08-03
> >
> > Thank you very much for your thorough and constructive feedback! We sincerely appreciate the time and insights into the review process, and we're glad to hear that our additional experiments and justifications were helpful.
> >
> > 1. **Benchmark Lifespan.** We agree this is an important point. We'll add a brief discussion in the introduction and conclusion about protecting the benchmark from saturation, including exploring appropriate licensing (e.g., CC-BY-NC) and holding back a portion of the test set. For future extensions of XIFBench, we plan to keep the statistically significant part of the data private to support long-term utility.
> >
> > 2. **Multilingual Model Analysis.** Thank you for highlighting this insight. We'll center our main discussion around these multilingual models, which we agree is important for understanding current model limitations in multilingual contexts.
> >
> > 3. **Carry-over Effect.** We appreciate your perspective here. We'll add a discussion on this potential effect, framing it as a broader methodological consideration for benchmarks that partially draw from existing datasets. We hope this will be useful for future work in this area.
> >
> > We'll incorporate these discussions, along with key points from our rebuttal, into the revised version. Thank you again for your thoughtful feedback.

---

### Official Review · Reviewer_Hbtn · 2025-06-27

**Rating:** 5
**Confidence:** 4

**Summary:**

This work introduces XIFBench, an evaluation benchmark for multilingual instruction-following. The benchmark includes 558 instructions translated to six languages ranging from high-resource (e.g. English) to low-resource (e.g. Swahili). The authors introduce a taxonomy of 5 constraint categories with 21 fine-grained dimensions; these span subjective categories (e.g. style) and objective ones (e.g. response length). Each instruction in the dataset is associated with up to 5 constraints. For evaluation, an LM judge is used to assess whether model responses follow the constraints in each instruction; the accuracy of the LM judge is verified against human annotators. The authors find that larger closed models outperform smaller open ones, and that smaller models show a steeper performance drop-off on low-resource languages. They also perform analysis to examine performance by constraint category and instruction complexity, finding that smaller models struggle with stylistic and situation constraints.

**Additional Feedback:**

- No line numbers in the preprint.
- More detail on the annotators would be useful. Are they paper authors, crowd workers, or some other group? How many were there, how much were they paid, etc.
- Section 3.1.3, paragraph on requirement decomposition: It was unclear to me why GPT was needed to do this decomposition given that you already have the brainstormed constraints available. You address this in Appendix C.3.1 but it might help to explain in the main paper body.
- Page 5, "Our evaluation confirms its high reliability": Give some numbers in the main paper body.
- Table 2: Showing averages by language for the closed and open models, respectively would be helpful to identify overall trends in language difficulty.
- Page 7, "Models like GPT-4o achieve RFR above 90% but IFR only around 70%, revealing limitations in full adherence": This is expected, right? If each individual constraint will be followed with prob. .9 and there are 5 constraints, then the probability that they'll all be followed is .9^5 = .6.
- Page 7, Direct Scoring: It wasn't totally clear to me how this works. This just ignores the fine-grained constraints entirely?

**Dataset Code Accessibility:**

Yes

**Dataset Code Comments:**

Data and code are available at the provided links.

**Ethical Considerations:**

No, there are no or only very minor ethics concerns

**Final Justification:**

My score hasn't changed from my original recommendation, which was to Accept.

**Limitations Weaknesses:**

I generally found the work to be methodologically sound and wasn't struck by any major weaknesses.

One point: in Sec. 4.1., GPT-4o is used as an LM judge. It could be helpful to check the stability of the results when using a different LM judge model. Additional minor points are included in the Additional Feedback section.

**Strengths Contributions:**

- Presents a novel benchmark to measure fine-grained instruction following across a range of languages, from high- to low-resource.
- Clearly written and provides a good level of detail on the data creation and evaluation process.
- Performs good validity checks, including:
  - Agreement of LM judge vs humans,
  - Accuracy of instruction translation / presence of "translationese"
  - Language used for evaluation requirements (compares always using English for evaluation vs. using the same language as the instruction).

---

> ### Author Rebuttal · Authors · 2025-07-31
>
> We sincerely appreciate your thorough and constructive feedback! Please find our responses to your valuable suggestions below.
>
> ---
>
> **LW1**: We agree that it's important to validate our evaluation with different LLM judges. To address this,
> we conducted additional experiments using a diverse set of models: GPT-4.1 (from the same family as GPT-4o), Gemini-2.5-Flash (a strong and cost-effective alternative), and DeepSeek-R1 (a leading open-source model). We evaluated their agreement with human judgments, as established in Section 4.2.
>
> The results are summarized below:
>
> | Model Judge | En | Zh-cn | Ru | Ar | Hi | Sw |
> |-------------------|------|--------|------|------|------|------|
> | GPT-4o | 95.9 | 96.7 | 95.0 | 93.0 | 95.5 | 92.5 |
> | GPT-4.1 | 91.9 | 91.6 | 87.4 | 89.1 | 90.5 | 84.6 |
> | Gemini-2.5-Flash | 90.2 | 91.9 | 85.3 | 87.0 | 89.5 | 86.0 |
> | DeepSeek-R1 | 90.9 | 90.2 | 80.0 | 82.8 | 84.2 | 87.0 |
>
> We observe that:
> - GPT-4o achieves the highest agreement with human judgments across all languages, which may be related to the evaluation prompt being iteratively refined with GPT-4o.
> - For high-resource languages (e.g., English, Chinese), different LLM judges yield broadly consistent results, indicating stable evaluation.
> - For mid- and low-resource languages, the variation across judges is more noticeable, suggesting room for further refinement in prompt design or language selection.
>
> We will include this table and the relevant discussion in the revised version of the paper.
>
> ---
>
> **AF1**: We apologize for the inconvenience. We followed the official NeurIPS 2025 Datasets and Benchmarks FAQ, which requires `\usepackage[preprint]{neurips_2025}` for single-blind submission, and this setting removes line numbers by default.
>
> ---
>
> **AF2**: The annotations were conducted by a group of 12 individuals, including the paper authors and undergraduate/master's students from collaborating universities. Some annotators contributed to multiple stages of the pipeline. Given the varying difficulty of tasks, non-author annotators were compensated based on estimated time effort, with a total cost of approximately $2800. We will clarify these details in the revised paper.
>
> ---
>
> **AF3**: We'll briefly clarify in Section 3.1.3 why we do requirement decomposition in our revised version.
>
> ---
>
> **AF4**: We'll report the proportion of acceptable translations for readability in Section 3.1.4 in our revised version.
>
> ---
>
> **AF5**: We'll add the average RFR and IFR by language for both closed- and open-source models in Table 2 in our revised version. For reference, the current averages are:
>
> | Language | En | Zh-cn | Ru | Ar | Hi | Sw |
> |----------|-------|-------|-------|-------|-------|-------|
> | **Closed-source Models** | | | | | | |
> | Avg. RFR | 92.0 | 88.9 | 90.1 | 86.2 | 87.0 | 84.9 |
> | Avg. IFR | 73.7 | 67.7 | 70.9 | 63.7 | 63.0 | 58.3 |
> | **Open-source Models** | | | | | | |
> | Avg. RFR | 89.0 | 85.8 | 85.4 | 77.1 | 71.8 | 35.3 |
> | Avg. IFR | 63.1 | 55.6 | 55.9 | 43.1 | 35.1 | 10.9 |
>
> ---
>
> **AF6**: Thank you for the insightful observation. We agree that a drop in IFR with increasing constraint count is expected. We also find that the rate of decline varies notably across models, which we believe provides useful signal about instruction-following robustness.
>
> We also analyzed this phenomenon in more detail (Appendix G.2). Interestingly, RFR (requirement following rate) remains relatively stable as constraint count increases, suggesting that models tend to follow a consistent fraction of constraints. We hope this helps address your concern.
>
> ---
>
> **AF7**: We apologize for the ambiguity. The *Direct Scoring* method follows the "single answer grading" setup described in Section 3.1 of [1]. Given a model response, the LLM judge provides an overall score based on dimensions such as helpfulness, relevance, accuracy, depth, creativity, and level of detail. This approach does not explicitly assess whether each constraint is satisfied, but rather offers a holistic judgment of the response quality.
>
> We'll refine the explanation in Section 4.2 (Agreement Evaluation) and Appendix F.2 to better clarify how *Direct Scoring* is conducted.
>
> [1] Judging LLM-as-a-Judge with MT-Bench and Chatbot Arena

---

> > ### Comment · Reviewer_Hbtn · 2025-08-01
> >
> > Thanks for the responses from the authors. They've addressed my questions and the additions they've included will clarify the final draft. I'm keeping my score as-is.

---

> > ### Author Response · Authors · 2025-08-02
> >
> > Thank you again for your thoughtful feedback! We're glad to hear that our responses addressed your questions, and we will incorporate the clarifications into the revised version to strengthen the presentation and enhance the impact of the work.

---

### Official Review · Reviewer_KYwr · 2025-06-29

**Rating:** 5
**Confidence:** 3

**Summary:**

Large Language Models (LLMs), like ChatGPT or GPT-4, are good at following instructions. But most tests focus only on English. This paper asks: *Can these models follow instructions just as well in other languages — like Swahili, Hindi, Arabic, Russian, or Chinese?*

To answer that, the authors created a big test set called XIFBench. It includes 558 instructions written in English and translated into 5 more languages. Three methodological innovations are offered: (i) cultural‑accessibility annotation, (ii) constraint‑level translation validation, and (iii) requirement‑based evaluation adapted to multilingual settings. They tested 9 different LLMs (both open and closed source) using XIFBench. Each model had to follow instructions in 6 languages, and they checked whether the model:

1. Followed each individual rule (called Requirement Following Rate, or RFR)
2. Followed all the rules together (called Instruction Following Rate, or IFR)

Experiments with several LLMs reveal how language‑resource level, constraint type, instruction complexity and cultural specificity affect adherence rates.

**Additional Feedback:**

A new benchmark called M-IFEval was released around early 2025, extending IFEval to French, Japanese, and Spanish, and mFollowIR focuses on retrieval instructions in Russian, Chinese, and Persian. Worth to mention in the paper.

- https://arxiv.org/abs/2502.04688
- https://arxiv.org/abs/2501.19264

**Dataset Code Accessibility:**

Partly

**Dataset Code Comments:**

The paper states the benchmark “will be released”, but the repository link is missing at submission time.

**Ethical Considerations:**

No, there are no or only very minor ethics concerns

**Final Justification:**

The rebuttal has sufficiently addressed my concerns. I have no further questions and will keep my Accept recommendation.

**Limitations Weaknesses:**

1. Reliance on Google‑Translate → GPT‑4o back‑validation. This may propagate translation artifacts; only 10 % was human‑audited. A larger human sample or bilingual reviewers would strengthen confidence. Another way of validation is consistency across multiple LLM models. And give the most inconsistent data to human reviewers.

2. Evaluation uses LLM judges exclusively. No human rating of model outputs is reported; this risks circularity.

3. Synthetic constraints only. All additional constraints are GPT‑generated; real user instructions (e.g., customer‑service logs) would improve ecological validity.

4. Cultural annotations (i.e., whether an instruction includes culturally specific references) are applied only to the English source. These labels are then assumed to hold across all translations—ignoring the fact that a neutral English instruction might become culturally specific in Arabic or Hindi due to translation choices or regional context.

**Strengths Contributions:**

+ Novel cross‑lingual scope. Prior constraint‑benchmarks focus on English/Chinese; XIFBench covers six typologically diverse languages and explicitly includes a low‑resource pair (Swahili).
+ Fine‑grained requirement extraction. The paper decomposes each constraint into atomic yes/no checks, enabling interpretable error profiles and radar‑plot analysis.
+ Cultural accessibility metric. Annotators label whether a constraint requires culture‑specific knowledge, yielding new insights (e.g., IFR more sensitive than RFR).
+ The paper compares with several major English or Chinese constraint benchmarks: CELLO, CoDI-Eval, IFEval, FollowBench, InfoBench, CFBench, ComplexBench—explaining that none of them cover multilingual constraint evaluation

---

> ### Author Rebuttal · Authors · 2025-07-31
>
> We sincerely appreciate your perceptive and insightful feedback! Please find our responses to your valuable suggestions below.
>
> ---
>
> **LW1**: Thank you for your helpful suggestions. In response, we have expanded our human auditing process as follows:
>
> - We increased the human audit coverage from 10% (10 Easy + 50 Hard English instructions, with their translations) to 20% (18 Easy + 90 Hard).
> - Instead of one annotator per language, each language is now evaluated by two native-speaking annotators who score independently.
> - We report the lower of the two scores for each translation to conservatively estimate translation quality.
>
> The updated translation quality scores (on a 1–5 scale) are summarized in the table below. Over 97.7% of translations received a score ≥ 3, indicating good quality. These results will be included in Appendix C.4 of the revised version.
>
> | Score | Zh-cn | Ru | Ar | Hi | Sw |
> |-------|-------|------|------|------|------|
> | 5 | 81.5 | 79.5 | 83.0 | 76.8 | 79.3 |
> | 4 | 13.3 | 15.9 | 12.0 | 17.8 | 15.9 |
> | 3 | 4.2 | 3.3 | 2.7 | 4.3 | 3.7 |
> | 2 | 1.0 | 0.8 | 2.3 | 0.8 | 1.0 |
> | 1 | 0.0 | 0.4 | 0.0 | 0.2 | 0.0 |
>
> ---
>
> **LW2**: To address this concern, we'd like to clarify that we conducted human evaluation for 1080 *instruction-response* pairs across six languages during the *Agreement Evaluation* stage (Section 4.2). We report the human-annotated RFR (requirement following rate) and IFR (instruction following rate) as below:
>
> **RFR:**
>
> | Model | En | Zh-cn | Ru | Ar | Hi | Sw |
> |------------------------|------|-------|------|------|------|------|
> | Gemini-2.0-Flash | 90.6 | 90.3 | 89.5 | 88.9 | 89.5 | 88.1 |
> | Qwen-2.5-72B | 92.7 | 86.3 | 85.9 | 82.1 | 76.8 | 45.3 |
> | Glm-4-9B | 83.2 | 80.0 | 74.4 | 71.7 | 62.5 | 25.6 |
>
> **IFR:**
>
> | Model | En | Zh-cn | Ru | Ar | Hi | Sw |
> |------------------------|------|-------|------|------|------|------|
> | Gemini-2.0-Flash | 73.3 | 68.3 | 66.6 | 61.7 | 66.6 | 60.0 |
> | Qwen-2.5-72B | 71.6 | 60.0 | 51.6 | 50.0 | 43.3 | 5.0 |
> | Glm-4-9B | 50.0 | 48.3 | 36.6 | 28.3 | 23.3 | 3.3 |
>
> These results provide a human-grounded view of model behavior. We will include them in Appendix F.2 in the revised version.
>
> ---
>
> **LW3**: Incorporating real user instructions is indeed a valuable direction for enhancing both realism
> and ecological validity, and we fully support this perspective.
>
> Given the resource constraints in academic (e.g., limited annotation budget), collecting real user instructions at scale remains challenging. As a practical compromise, we designed our constraint taxonomy by drawing on prior work such as CFBench [1], and by synthesizing common constraint patterns observed in benchmarks like AlpacaEval [2]. We believe this helps approximate realistic scenarios while maintaining scalability.
>
> To assess the reliability of GPT-generated constraints, we revisited the instruction validation annotations in Section 3.1.2. Among the 18 removed instructions (3.4%) that lacked clarity, only 2 were due to inappropriate constraints. For example, "use a problem-solving framework to discuss dark adaptation challenges" appeared somewhat misaligned with the base instruction "How long does it take our eyes to fully adapt to darkness?". This suggests that GPT-generated constraints are generally appropriate.
>
> While synthetic constraints offer a controlled starting point, we recognize the value of grounding benchmarks in real-world usage. We plan to incorporate instructions and constraints from real-world datasets such as WildChat [3], which we view as a critical step in advancing the benchmark’s realism and applicability.
>
> [1] CFBench: A Comprehensive Constraints-Following Benchmark for LLMs
> [2] AlpacaEval: An automatic evaluator of instruction-following models.
> [3] WildChat: 1M ChatGPT Interaction Logs in the Wild
>
> ---
>
> **LW4**: The reviewer's observation highlights a subtle yet important point, which we fully acknowledge and appreciate.
>
> To mitigate this risk, our annotation of cultural accessibility was performed with awareness that the dataset would be translated into multiple languages. Annotators were instructed to focus on universally understandable content and avoid cultural references involving geographic, linguistic, or cultural background knowledge. See Appendix C.1.2 and Table 4 for guideline details.
>
> We fully agree that cultural specificity would emerge during translation due to translation choices or regional context. We are actively analyzing cultural drift across translations and plan to include the findings in the revised version, subject to timeline constraints.
>
> ---
>
> **DCC1**: We apologize for the non-clickable "will be released" footnote on Page 1 and will correct this in our revised version.
>
> In the meantime, the full dataset and code are already available through the anonymous GitHub link provided in the `Code URL` field on the OpenReview submission page. This is the same repository intended for public release.
>
> ---
>
> **AF1**: We will include both works in the Related Work section of our revised version. Specifically, we will mention that M-IFEval [1] extends IFEval to French, Japanese, and Spanish, and that mFollowIR [2] evaluates the instruction-following capabilities of retrieval models in Russian, Chinese, and Persian.
>
> [1] M-IFEval: Multilingual Instruction-Following Evaluation
> [2] mFollowIR: a Multilingual Benchmark for Instruction Following in Retrieval

---

> > ### Comment · Reviewer_KYwr · 2025-08-06
> >
> > The rebuttal has sufficiently addressed my concerns. I have no further questions and will keep my Accept recommendation.

---

> > > ### Author Response · Authors · 2025-08-06
> > >
> > > Thank you again for your thoughtful feedback! We're glad our responses addressed your concerns, and we'll incorporate the clarifications into the revised version.

---

### Official Review · Reviewer_Jvrs · 2025-07-03

**Rating:** 5
**Confidence:** 3

**Summary:**

This work proposes a new benchmark dataset for large language models (LLMs), XIFBench, to solve the problem caused by conventional constrained instruction-following benchmarks that only cover high-resource languages like English and Chinese. In contrast to the conventional benchmark datasets, the proposed XIFBench covers Chinese, Russian, Arabic, Hindi, and Swahili. Furthermore, XIFBench comprises systematically augmented instructions, with 0 to 5 constraints drawn from a taxonomy covering five categories (Content, Style, Situation, Format, and Numerical) and 21 fine-grained dimensions. The experimental results of 9 LLMs on XIFBench show that these are weak in non-English languages. In addition, the authors analyze the results and reveal that some strong LLMs are still weak in non-English languages and do not cover cultural aspects.

**Additional Feedback:**

How did the translation change the instruction length? It is an essential factor to consider the performance changes across languages.

**Dataset Code Accessibility:**

Yes

**Dataset Code Comments:**

We can access their created instruction data, which accompanies a short description.

**Ethical Considerations:**

No, there are no or only very minor ethics concerns

**Final Justification:**

Since my major concern about the evaluation bias had been solved by the authors' response, I have raised my score.

**Limitations Weaknesses:**

- The dataset expansion for covering multiple languages relies on machine translation. It may cause a lack of cultural aspects and topics.
- In the automatic evaluation, only one LLM, GPT-4o, is used as the evaluator. It may cause a bias in the evaluation results.

**Strengths Contributions:**

- The newly created XIFBench covers non-centric languages.
- Not only languages but also cultural aspects are covered by XIFBench
- Human annotators verify the created datasets, and this process ensures the reliability of XIFBench
- The authors conduct a comparison of LLM performance on culturally universal and specific
instructions.
- The translated instructions are also verified by human annotators.
- Even though the automatic evaluation is based on the LLM-as-a-judge manner, the authors check the correlation between the performance from this approach and humans, and readers can understand the validity of the evaluation.

---

> ### Author Rebuttal · Authors · 2025-07-31
>
> We sincerely appreciate your thoughtful and insightful feedback! Please find our responses to your valuable suggestions below.
>
> ---
>
> **LW1**: We appreciate your attention to the cultural and topical limitations. As you noted, translation offers a practical first step toward multilingual expansion, but it may introduce English-centric biases—a limitation we've tried to be mindful of in the paper, and one we see as important for future improvement.
>
>
> To better understand its impact, we revisited our cultural accessibility annotations and found a few non-English-centric examples (e.g., "say 'How are you' in Korean"), though such cases are relatively rare. We agree that incorporating annotations for non-English instructions could help improve cultural coverage. We hope our current release can serve as a starting point toward that goal.
>
> ---
>
> **LW2**: We appreciate your concern regarding potential bias from relying on a single evaluator. To address this, we conducted additional experiments using a diverse set of state-of-the-art LLMs and examined their agreement with human judgments.
>
> Specifically, we evaluated the following models:
> - **GPT-4.1** — to assess consistency within the same model family as GPT-4o.
> - **Gemini-2.5-Flash** — a strong, cost-efficient model from a different family.
> - **DeepSeek-R1** — a leading open-source model with strong reasoning capabilities.
>
> We used the human annotations from Section 4.2 ("Agreement Evaluation") as ground truth. The table below summarizes the human agreement rates. GPT-4o is the original evaluator used in our paper.
>
> | Model Judge | En | Zh-cn | Ru | Ar | Hi | Sw |
> |-------------------|------|--------|------|------|------|------|
> | GPT-4o | 95.9 | 96.7 | 95.0 | 93.0 | 95.5 | 92.5 |
> | GPT-4.1 | 91.9 | 91.6 | 87.4 | 89.1 | 90.5 | 84.6 |
> | Gemini-2.5-Flash | 90.2 | 91.9 | 85.3 | 87.0 | 89.5 | 86.0 |
> | DeepSeek-R1 | 90.9 | 90.2 | 80.0 | 82.8 | 84.2 | 87.0 |
>
> We observe that:
> - **GPT-4o** achieves the highest agreement with human judgments across all languages, which may be related to the evaluation prompt being iteratively refined with GPT-4o.
> - **GPT-4.1**, being from the same model family, shows the most consistent performance among the alternatives.
> - For high-resource languages (e.g., English, Chinese), all LLM judges perform comparably. For mid- and low-resource languages (e.g., Russian, Arabic, Hindi), performance gaps become more significant across models.
>
> These results help validate the robustness of our evaluation setup and offer a broader view of evaluator reliability across different models. We will incorporate this table and the corresponding discussion into the revised paper.
>
> ---
>
> **AF1**: We agree that instruction length is an important factor in understanding cross-lingual performance differences. To study this, we conducted additional experiments.
>
>
> We measured instruction length (in tokens) using the `tiktoken` library with the `o200k_base` tokenizer (used by GPT-4o) over the instruction part of our XIFBench dataset (excluding *input materials*, which are not translated). We then computed the Pearson correlation between instruction length and the average RFR (Requirement Following Rate) and IFR (Instruction Following Rate) across both closed-source and open-source models.
>
> The results are summarized below:
>
> | Language | Token Length (Mean ± Std.) | RFR Avg. (Closed Source) | IFR Avg. (Closed Source) | RFR Avg. (Open Source) | IFR Avg. (Open Source) |
> | :------- | :------------------------- | :----------------------- | :----------------------- | :--------------------- | :--------------------- |
> | En | 47.8 ± 22.2 | 92.0 | 73.7 | 88.9 | 63.1 |
> | Zh-cn | 53.2 ± 25.0 | 88.9 | 67.7 | 85.8 | 55.6 |
> | Ru | 68.4 ± 32.0 | 90.1 | 70.9 | 85.4 | 55.6 |
> | Ar | 67.6 ± 31.6 | 86.2 | 63.7 | 77.1 | 43.1 |
> | Hi | 82.2 ± 38.9 | 87.0 | 63.0 | 71.8 | 35.1 |
> | Sw | 78.5 ± 37.1 | 84.9 | 58.3 | 35.3 | 10.9 |
> | **Pearson Corr.** | - | -0.77 | -0.78 | -0.67 | -0.79 |
>
> Our key observations are:
> - **Translating from English to other languages generally increases instruction length.** We observe a step-like pattern: the gap in token length is more prominent between high-/medium-/low-resource languages (e.g., English/Chinese vs. Russian/Arabic vs. Hindi/Swahili) than within each tier.
> - **There is a clear negative correlation between instruction length and model performance** across both model types and metrics. Longer instructions tend to result in lower RFR and IFR scores.
> - We observed that Ru/Ar and Hi/Sw have similar token lengths but show notable RFR differences. Conversely, Zh-cn/Ru and Ar/Hi have larger length differences but smaller performance gaps. This suggests that other factors beyond instruction length may also influence cross-lingual performance.
>
> We will include this analysis in the appendix of our revised paper.

---

> > ### Comment · Reviewer_Jvrs · 2025-08-05
> > **The response satisfies my requirement**
> >
> > I appreciate the detailed response from the authors. Since my major concern about the evaluation bias is solved by the response, I will raise my score. I believe that the authors write the left issues in the limitations or future work sections in the camera-ready version of their paper.

---

> > > ### Author Response · Authors · 2025-08-06
> > >
> > > Thank you again for your thoughtful feedback! We're glad to hear that our responses addressed your main concern. We will incorporate the remaining issues and additional experiments into the revised version to improve the coverage and completeness of the work.

---

### Decision · Program_Chairs · 2025-09-18

**Decision:**

Accept (poster)

**Comment:**

All reviewers have voted to accept the paper. The authors have angaged in discussions with the reviewers and added new experiments and results.